# An experimental study on light scattering matrices for Chinese loess dust with different particle size distributions

Jia Liu[1], Qixing Zhang[1], Yinuo Huo[1], Jinjun Wang[1], Yongming Zhang[1]

[1]State Key Laboratory of Fire Science, University of Science and Technology of China, Hefei, Anhui, 230026, China

*Correspondence to*: Qixing Zhang (qixing@ustc.edu.cn) and Yongming Zhang (zhangym@ustc.edu.cn)

**Abstract**

Mineral dust suspended in the atmosphere has significant effects on radiative balance and climate change. Chinese Loess Plateau (CLP) is generally considered as a main source of Asian dust aerosol. After being lifted by wind, dust particles with various size distributions can be transported over different distances. In this study, original loess sample was collected from

Luochuan, which is centrally located at CLP, and two samples with different size distributions were obtained after then. "Pristine loess" was used to represent dust that only affect source regions, part of "pristine loess" was milled to finer "milled loess" that can be transported for long distance. Light scattering matrices for these two samples were measured at 532 nm wavelength from 5 ° to 175 ° angles. Particle size distribution, refractive index, chemical component, and microscopic appearance were also characterized for auxiliary analyses. Experimental results showed that there are obvious discrepancies

in angular behaviours of matrix elements for "pristine loess" and "milled loess", and these discrepancies are different from that for other kinds of dust with distinct size distributions. Given that the effective radii of these two loess samples differ by more than 20 times, it is reasonable to conclude that the difference in size distributions plays a major role in leading to different matrices, while differences in refractive index and micro structure have relatively small contributions. Qualitative analyses of numerical simulation results of irregular particles also validate this conclusion. Gaussian spheres may be

promising morphological models for simulating scattering matrix of loess but need further quantitative verification. At last, synthetic scattering matrices for both "pristine loess" and "milled loess" were constructed over 0 °-180 °, and the previous average scattering matrix for loess dust was updated. This study presents measurement results of Chinese loess dust and updated average scattering matrix for loess, which are useful for validating existing models and developing more advanced models for optical simulations of loess dust and finally help to improve retrieval accuracy of dust aerosol properties over

both source and downwind areas.

## 1 Introduction

Mineral dust is a common particulate type in Earth's atmosphere, and accounts for a high fraction of atmospheric aerosol mass loading (Tegen and Fung, 1995). Asian dust contributes a lot to global atmospheric mineral dust aerosol, dust emitted from East Asia only is about $1.04 \times 10^7$ ton/year, $2.76 \times 10^7$ ton/year and $5.13 \times 10^7$ ton/year for $PM_{10}$ (particles with

aerodynamic equivalent diameter smaller than 10 μm), PM30 and PM50 (Xuan et al., 2004). During aerosol characterization experiments ACE-Asia, mass balance calculations indicated that 45-82 % of atmospheric aerosol mass at observation sites in China were attributed to Asian dust (Zhang et al., 2003). Chinese Loess Plateau (CLP) is usually considered as a main source or an important supply site of Asian dust aerosol (Han et al., 2008; Shen et al., 2016; Tsai et al., 2014; Zhang et al., 2010). Statistical analysis of dust storms influencing Chinese Mainland from 2000 to 2002 showed that about a quarter of dust

storms were originated from CLP (Zhang and Gao, 2007). Source tracing of dust collected in Xi'an city revealed that these dust particles were mainly short-distance transported from CLP (Yan et al., 2015). Comparisons of chemical element ratios demonstrated that dust particles emitted from CLP can be transported to Korea, Japan and North Pacific (Cao et al., 2008).

      Because of the scattering and absorption of solar radiation, atmospheric dust has remarkable influences on global climate change as well as radiation budget (Satheesh and Moorthy, 2005; Sokolik and Toon, 1996). Dust particles with

different sizes can be transported over different distances, more specifically, dust particles with a size range of r > 5 μm exist in source areas only, while particles with a size range of 0.1 < r < 5 μm can experience airborne transportation over long distances (like about 5000 km), even cross-continent from Asia to North America (Jaffe et al., 1999; Satheesh and Moorthy, 2005). Therefore, loess dust emitted from CLP is expected to have important influence on the radiation balance at both source areas and places far away from sources.

It is well known that dust particles have distinct non-spherical shapes, thus retrievals of dust aerosol properties, like optical thickness, based on Lorenz-Mie computations will lead to significant errors (Herman et al., 2005; Mishchenko et al., 2003). Optical modeling of dust particles with non-spherical shapes has been an essential subject. Dubovik et al. (2006) employed a mixture of spheroids with different axial ratios as well as spheres to reproduce laboratory measured angular light scattering patterns of dust aerosols presented by Volten et al. (2001), and the best fitted shape distribution of spheroids was

obtained and proposed. Subsequent studies on the retrievals of dust aerosol properties from space-based (Dubovik et al., 2011), airborne (Espinosa et al., 2019) and ground-based (Titos et al., 2019) remote sensing observations were all based on this shape distribution. However, the application of a same shape distribution of spheroids for different kinds of dust is somewhat too arbitrary (Li et al., 2019) and may not be suitable for simulating optical properties of loess dust with different size distributions. Furthermore, more precise optical models which are more complex than spheroids and similar to real dust

morphology are still needed. Laboratory measurements of angular scattering patterns as well as basic physical features, like size distribution, refractive index and micro structure, of loess dust with different sizes are essential and beneficial to the development of more precise models for loess dust. These models will further useful for more accurate retrievals of dust aerosol properties over both source and downwind regions from remote sensing observations, and more accurate assessments of radiative forcing at different regions.

Optical properties of dust particles vary with changes of their size distributions. Light scattering matrix $F$, a 4×4 matrix containing 16 elements $F_{ij}$ ($i$, $j$=1-4), is a fundamental optical property to characterize airborne dust particles, and describes the depolarization or transformation of incident light with several polarization states under the influences of particles (Quinby-Hurt et al., 2000; Volten et al., 2001). Scattering matrix is not only sensitive to size distribution but also sensitive to

physical features like particle shape, micro structure and refractive index (Muñoz and Hovenier, 2011). Therefore, it can be employed as a useful parameter to provide information and implications about above features of dust particles. Based on similar operational principles, several light scattering matrix measurement apparatuses were developed by researchers in the past two decades (Liu et al., 2018; Muñoz et al., 2010; Volten et al., 2001; Wang et al., 2015). With the assistant of these apparatuses, scattering matrices for various mineral dust were experimentally determined, such as loess, clay, desert dust, volcanic ash, simulant of cosmic dust and so on (Dabrowska et al., 2015; Escobar-Cerezo et al., 2018; Merikallio et al., 2015; Muñoz et al., 2007; Muñoz et al., 2001). In addition, Amsterdam Database and Amsterdam-Granada Database were established at 2005 and 2012 to publish measured scattering matrices as well as necessary physical properties of mineral dust particles (Volten et al., 2005; Volten et al., 2006a; Muñoz et al., 2012).

Most published literatures of experimental measurements of scattering matrices focused more on similarities and discrepancies between different kinds of mineral dust, or between the same kinds of dust sampled from different sources. Furthermore, some researches paid more attention to the effect of particle size distribution on scattering matrices. Olivine dust with four size distributions were obtained using different sieves, but there are no clear and consistent effects of size on measured scattering matrices for olivine at both 442 nm and 633 nm wavelengths (Muñoz et al., 2000). Forsterite samples were produced with three size distributions using dry and wet sieving methods, comparisons of experimental scattering matrices at 632.8 nm wavelength clearly showed the influence of size (Volten et al., 2006b). Relative phase function is larger for large forsterite particles, $F_{22}/F_{11}$ is larger for small particles, $-F_{12}/F_{11}$ and $F_{34}/F_{11}$ for small particles are larger at most scattering angles but there are opposite trends for the negative branches at backscattering angles, $F_{33}/F_{11}$ and $F_{44}/F_{11}$ for small particles are larger at forward scattering angles while are smaller at backscattering angles. Two samples of palagonite with different size distributions were prepared by heating, analyses of measured $-F_{12}/F_{11}$ revealed that small particles have larger $-F_{12}/F_{11}$ values at both 488 nm and 647 nm wavelengths (Dabrowska et al., 2015). Three commercial samples of Arizona Road Dust consisting of ultrafine, fine and medium particles were selected to investigate their scattering matrices, results demonstrated that ultrafine particles have the largest normalized phase function while medium particles have the smallest $F_{22}/F_{11}$ (Wang et al., 2015). Lunar soil simulant JSC-1A particles were recovered and reused during scattering matrices measurement experiments, recovered sample was larger than pristine sample, comparative analyses indicated that large particles have larger relative phase function and $-F_{12}/F_{11}$, large particles have smaller $F_{22}/F_{11}$ at forward scattering angles while $F_{22}/F_{11}$ for these two samples were nearly consistent at backscattering angles (Escobar-Cerezo et al., 2018). Experimentally determined $-F_{12}/F_{11}$ for meteorites illustrated that the minimum value of $-F_{12}/F_{11}$ for larger particles is smaller, and the maximum value of $-F_{12}/F_{11}$ for larger particles is larger (Frattin et al., 2019).

It can be concluded from above researches that size distributions have inconsistent effects on scattering matrix elements for different kinds of dust particles. And there is no study pay attention to the effect of size distribution on scattering matrix for loess dust. Therefore, loess dust with different size distributions were investigated in this study. Original loess sample was collected from Luochuan, the center of CLP, after sieving to remove oversized particles, "pristine loess" sample was used to represent loess dust that is only present in source regions. Furthermore, part of "pristine loess"

was ball-milled to obtain finer "milled loess" sample that can be transported over long distance and affect regions far away from dust sources. Scattering matrices for above loess samples with distinct size distributions were measured at 532 nm with the help of a self-developed and validated apparatus over angles 5°-175°. Besides particle size distribution, other characteristics that might be changed during milling process were also analyzed, such as chemical component, refractive index and microscopic appearance. Discrepancies in angular behaviors of matrix elements were summarized and their reasons were discussed based on analyses of numerical simulations in literatures. Furthermore, synthetic scattering matrices were defined over 0°-180°, and the previously published average scattering matrix for loess was updated.

In Section 2, fundamental characteristics of "pristine loess" and "milled loess" samples are shown. In Section 3, concise descriptions of related theory, apparatus and methods are given. In Section 4, measured and synthetic scattering matrices for these two samples are plotted, reasons leading to these discrepancies in matrix elements are discussed and previous average scattering matrix for loess is updated. At last, in Section 5, conclusions are drawn.

## 2 Fundamental Characteristics of Loess Dust Samples

There are two deserts in the northern of Chinese Loess Plateau, and according to the distances from these deserts, CLP is roughly separated into 3 regions: sandy loess, loess as well as clayey loess (Cao et al., 2008). Original loess dust sample was collected from Loess National Geological Park (35.76 °N, 109.42 °E) at Luochuan, which is lying on "loess zone" and also at the center of CLP. Since this park is the only national geological park in China which has typical loess geomorphology, it can be considered that the sample collected represents Chinese loess to a certain extent. Prior to laboratory investigations, oversized particles in original sample were removed through a 50 μm sieve. Next, the original loess sample was divided into two parts, one of which was not treated any more and was called as "pristine loess", and the other was milled by a ball miller to obtain finer particles, called as "milled loess". It should be noted that "milled loess" is the same sample as the "Luochuan loess" in reference (Liu et al., 2019). Both of these two loess dust samples were investigated through light scattering matrices measurements as well as auxiliary analyses of other physical characteristics of particles.

The size distributions of "pristine loess" and "milled loess" were determined by a laser particle sizer (SALD-2300; Shimadzu) using dry measurement method, dry loess particles were injected into the measurement unit of laser particle sizer, and three independent repeated measurements were conducted for each sample. As can be seen from Figure 1, the size of "pristine loess" shows a distinct bimodal distribution, after ball milling, particle size of "milled loess" becomes a unimodal distribution. From the viewpoint of atmospheric particle transportation, the majority (number fraction more than 70%) of "pristine loess" particles have radii larger than 5 μm with peaks at about 3.9 and 10.7 μm, thus this sample can be used to represent coarse dust that only affect source regions, like Xi'an City (Yan et al., 2015). On the other hand, almost all particles of "milled loess" sample have radii smaller than 2 μm with a peak at about 0.55 μm, and can be used as a representative of fine dust that can be transported over long distance and affect regions far away from dust sources.

SALD-2300 has 84 scattering light detectors in all, including 78 forward detector elements, one side detector and five back detectors. The best fitted number size distribution and refractive index $m$ can be obtained by reproducing measured angular distribution of light intensity based on Mie calculations. Liu et al. (2003) revealed that Mie theory can be used to reproduce forward scattering intensities of nonspherical particles with moderate aspect ratios at scattering angles smaller than 20°. Since over 70% of the detectors of SALD-2300 are set at angles smaller than 20°, the retrieved size distributions of nonspherical loess dust based on Mie theory are of relatively high accuracy. During size distribution measurements of loess samples, the retrieval ranges of real part $Re(m)$ and imaginary part $Im(m)$ of refractive index were preset as 1.45-1.75 and 0-0.05, respectively (Volten et al., 2001). The smallest calculation steps of $Re(m)$ and $Im(m)$ are 0.05 and 0.01, respectively. As shown in Table 1, the optimal refractive indices are $1.65+0i$ for "pristine loess" and $1.70+0i$ for "milled loess", larger particles have relatively smaller real part of refractive index, which is similar to the results of Kinoshita (2001) and is caused by the nonspherical nature of loess dust. Retrieved refractive index of particles based on measured light intensity distribution is a kind of optically equivalent refractive index, which is close to the inherent refractive index of the measured particles. Based on measured size distributions, effective radius $r_{eff}$ as well as standard deviation $\sigma_{eff}$ can be derived (Hansen and Travis, 1974):

$$r_{eff} = \frac{\int_0^\infty r\pi r^2 n(r)dr}{\int_0^\infty \pi r^2 n(r)dr} \tag{1}$$

$$\sigma_{eff} = \sqrt{\frac{\int_0^\infty (r-r_{eff})^2 \pi r^2 n(r)dr}{r_{eff}^2 \int_0^\infty \pi r^2 n(r)dr}} \tag{2}$$

where $n(r)dr$ stands for number proportion of equivalent spheres whose radii vary between $r$ and $r+dr$. Results of $r_{eff}$ and $\sigma_{eff}$ are shown in Table 1. In addition, effective size parameters $x_{eff} = 2\pi r_{eff}/\lambda$ for "pristine loess" and "milled loess" were also calculated and presented in Table 1.

**Figure 1**

**Table 1**

Scanning electron microscope (SEM) images for "pristine loess" (left panel) and "milled loess" (right panel) are displayed in Figure 2. Obviously, particles of these two samples exhibit various shapes, and all of the particles can be classified as irregular shape. Almost all particles have rough surfaces and some particles even have sharp edges. After the milling process, there are more sub-micron particles in "milled loess" sample, some small particles even stuck on the rough surfaces of large particles due to electrostatic forces.

**Figure 2**

During the dry milling process, non-metal grinding balls with 6 mm diameter were used, the main component of which is $ZrO_2$. For the purpose of detecting whether the chemical compositions of loess samples were changed, the oxide compositions of samples before and after milling process, that is the "pristine loess" and "milled loess", were determined using a X-ray fluorescence spectrometer (XRF-1800, Shimadzu), the detection limit of which is 0.0001 wt %. As can be seen

 that the composition differences between these two samples are very small, and milling process has little effect on chemical compositions for loess samples.

**Table 2**

## 3 Theoretical Background and Experimental Methodology

### 3.1 Basic Concepts about Light Scattering Matrix

Four Stokes parameters (*I*, *Q*, *U* and *V*) are usually used to introduce the intensity and polarization properties of light beam, and these parameters can form a column vector, the so called Stokes vectors (Hovenier et al., 2014; Hulst and Van De Hulst, 1981). If a cloud of particles present in light path, the incident beam will be scattered and part of light will deviate from the original direction of propagation. When multi-scattering plays a negligible role, intensity and polarization state of scattered beams can be calculated from that of incident beam, using a 4×4 light scattering matrix *F* (Mishchenko and Yurkin, 2017):

$$\begin{pmatrix} I_{sca} \\ Q_{sca} \\ U_{sca} \\ V_{sca} \end{pmatrix} = \frac{\lambda^2}{4\pi^2 D^2} \begin{pmatrix} F_{11}(\theta) & F_{12}(\theta) & F_{13}(\theta) & F_{14}(\theta) \\ F_{21}(\theta) & F_{22}(\theta) & F_{23}(\theta) & F_{24}(\theta) \\ F_{31}(\theta) & F_{32}(\theta) & F_{33}(\theta) & F_{34}(\theta) \\ F_{41}(\theta) & F_{42}(\theta) & F_{43}(\theta) & F_{44}(\theta) \end{pmatrix} \begin{pmatrix} I_{inc} \\ Q_{inc} \\ U_{inc} \\ V_{inc} \end{pmatrix} \tag{3}$$

where $\lambda$ stands for wavelength of light, *D* is the distances between particle cloud and light detectors, scattering angle $\theta$ is the angle between incident and scattered beams, and the scattering plane contains both incident and scattered beams.

Generally, *F* has 16 independent matrix elements $F_{ij}$ with *i*, *j*=1-4. Two basic assumptions are commonly used to simplify the general form of light scattering matrix. The first one is that all scattering planes are equivalent for particles having random orientations. Thus, scattering directions can be adequately depicted by $\theta$. The second one is that particles and their mirror counterparts exist in the same number in a cloud of randomly oriented particles. Based on above random orientation and mirror particle assumptions, the number of independent elements in light scattering matrix can be reduced from 16 to 6 (Mishchenko and Yurkin, 2017):

$$F = \begin{pmatrix} F_{11}(\theta) & F_{12}(\theta) & 0 & 0 \\ F_{12}(\theta) & F_{22}(\theta) & 0 & 0 \\ 0 & 0 & F_{33}(\theta) & F_{34}(\theta) \\ 0 & 0 & -F_{34}(\theta) & F_{44}(\theta) \end{pmatrix} \tag{4}$$

Matrix elements describe the depolarization or transformation of incident light with several polarization state under the influence of particles (Quinby-Hurt et al., 2000). $F_{11}$ describes transformation of incident light intensity; $F_{12}$ describes depolarization of 0° and 90° linearly polarized light relative to scattering plane; $F_{22}$ describes transformation of ±90°

polarized incident light to $\pm 90°$ polarized scattered light and it equals to $F_{11}$ for spherical particles; $F_{33}$ and $F_{44}$ describe transformation of $\pm 45°$ linearly (or circularly) polarized incident light to $\pm 45°$ linearly (or circularly) polarized scattered light and these two elements are equal for spherical particles; $F_{34}$ describes transformation of circularly polarized incident light to $\pm 45°$ linearly polarized scattered light. Almost all these matrix elements are sensitive to physical properties of particles, including size distribution, particle shape, micro structure and refractive index.

## 3.2 Experimental Apparatus and Methodology

Figure 3 shows a layout diagram of the improved scattering matrix measurement apparatus. The main improvement is that angle coverage at backscattering angles are extended to $175°$, while the maximum coverage of previous apparatus is $160°$ (Liu et al., 2018). The wavelength of incident beam is 532 nm, and there are a linear polarizer $P$ as well as an electro-optic modulator $EOM$ in its propagation path. Subsequently, the modulated incident light is scattered by particles in the scattering zone, which are dispersed using an aerosol generator and are sprayed upwards to scattering zone through a nozzle. A photomultiplier named as the "detector", a 532 nm quarter-wave plate $Q$ as well as a polarizer $A$ are fixed on a rotation arm, rotation center of which is coincides with the center of aerosol nozzle. Before scattered light is detected by the "detector", it successively passes through $Q$ and $A$. The dark cassette used to encapsulate the "detector", $Q$ and $A$ in previous apparatus is removed, which facilitate the adjustment of orientation angles of $Q$ and $A$. The "detector" is controlled by an electric rotary table and is able to scan scattering angles from $5°$ to $175°$. Another photomultiplier named as the "monitor" is fixed at $30°$ scattering angle to record variations of dust aerosols. The combination of electro-optic modulator and lock-in detector allows multiple scattering matrix elements or their sums can be measured simultaneously. All the matrix elements of dust samples can be determined as functions of scattering angles with the help of various combinations of orientation angles of above optical elements as shown in Table 3, which is just the same as Muñoz et al. (2010).

Figure 3

Table 3

Multiple groups of values of measurable quantities, that is the DC component $DC(\theta)$, first harmonics $S(\theta)$ and second harmonics $C(\theta)$ of voltage signal, are recorded at every scattering angle for each combination of optical elements. The first step of data processing is to average these recorded values and get their errors. The optical platform is surrounded by black curtains to avoid the effect of environmental stray light, and background signals need to be measured and subtracted. Fluctuations of dust aerosols can be eliminated by normalizing measurements of the "detector" using $DC(30°)$ measured by the "monitor". Scattering matrix elements can be extracted from preprocessed $DC(\theta)$, $S(\theta)$ and $C(\theta)$ according to Table 3. Subsequently, $F_{11}(\theta)$ is normalized to 1 at $10°$ scattering angle, and the remaining matrix elements $F_{ij}(\theta)$ are normalized to $F_{11}(\theta)$ at the same angle. At last, whether measurement results of scattering matrix satisfy Cloude coherency matrix test should be examined (Hovenier and Van Der Mee, 1996). Three iterations of measurements are performed for each particle sample, the final results are average of three groups of experiments, and the errors are also calculated which contain errors during every measurement and errors for repeat measurements. Furthermore, the improved apparatus is validated using water

droplets. Measured all six non-zero scattering matrix elements for water droplets can be well fitted using Mie calculation results, indicating that the measurement accuracy of apparatus are satisfactory. For more details about the measurement principle and validation method of the apparatus, it can be referred to Liu et al. (2018).

A dust generator (RBG 1000; Palas) was applied to disperse loess particles (Liu et al., 2018). Re-aerosolized dust aerosols were transported to scattering matrix measurement apparatus using conductive tube and sprayed upwards to scattering zone through nozzle. Some particles of each loess sample were sprayed into vessels or sprayed onto copper grids for subsequent size distribution measurements or SEM analyses. For reliable measurements of scattering matrix, experiments should be conducted under single scattering conditions. This requires that the number of particles in the scattering zone is appropriate, too many particles will result in significant multiple scattering, while too few particles will dissatisfy the two basic assumptions mentioned above. Incident light intensity $I_0$ as well as transmitted light intensity $I$ passing through particle cloud can be related by the following equation (Mokhtari et al., 2005):

$$I = I_0 e^{-\langle s \rangle} \tag{5}$$

where <s> stands for average number of scattering events. P(2)/P(1)=<s>/2 is used to describe the ratio of occurrence probability of double scattering event (the simplest form of multi-scattering) to that of single scattering event (Wang et al., 2015).

## 4 Results and Discussions

### 4.1 Experimentally Determined Scattering Matrices

The measurements of <s>/2 were conducted before the measurements of matrix elements using each orientation angle combination of above optical elements. Measured <s>/2 for both "pristine loess" and "milled loess" were smaller than about 0.006, in other word, the occurrence probability of double scattering event was about 170 times smaller than that of single scattering event and double scattering event can be ignored without question. For each loess sample, three independent and replicated measurements of scattering matrix were conducted, and experimental results shown in figures are averaged values for three measurements. Examinations showed that measurements of loess samples satisfy Cloude coherency matrix test at all scanned scattering angles.

Experimentally determined scattering matrix elements for both "pristine loess" and "milled loess" are shown in Figure 4. Only six element ratios are plotted, because other ratios do not deviate from zero within experimental errors. Matrix element ratios for "pristine loess" and "milled loess" present similar angular behaviors, more specifically, angular distributions of all six non-zero matrix element ratios are limited to narrow regions, respectively. Normalized phase functions $F_{11}(\theta)/F_{11}(10°)$ show strong forward scattering peaks, variations at backscattering directions are not obvious, which are typical behaviors for mineral dust with irregular shapes (Muñoz et al., 2012; Volten et al., 2001). For non-polarized incident beam, $-F_{12}(\theta)/F_{11}(\theta)$ is equivalent to the degree of linear polarization. Measured angular behaviors of $-F_{12}(\theta)/F_{11}(\theta)$ are bell-shaped, and the

largest values appear at near side-scattering directions. There are negative branches of $-F_{12}(\theta)/F_{11}(\theta)$ at both forward and backward scattering directions. $F_{22}(\theta)/F_{11}(\theta)$ is a proof of the non-sphericity and irregularity of particles, since it is constant 1 for homogeneous spheres. Measured values of these two loess samples show that $F_{22}(\theta)/F_{11}(\theta)$ values deviate from constant 1 at nearly all angles scanned. The ratios $F_{33}(\theta)/F_{11}(\theta)$ and $F_{44}(\theta)/F_{11}(\theta)$ can be analyzed jointly because these two ratios are equal for particles with spherical shape. But for loess dust, $F_{33}(\theta)/F_{11}(\theta)$ values are smaller than $F_{44}(\theta)/F_{11}(\theta)$, especially at backscattering directions. The ratios $F_{34}(\theta)/F_{11}(\theta)$ show near "S-type" shapes and the maximums are obtained at about 115 ° angle. For scattering angles smaller than 50 ° and larger than 170 °, values of $F_{34}(\theta)/F_{11}(\theta)$ are negative.

**Figure 4**

On the other hand, the discrepancies in matrix elements for "pristine loess" and "milled loess" are still obvious. Compared to "milled loess", there is an enlargement of relative phase function at 5° scattering angle for "pristine loess". Relative phase function for "pristine loess" is also larger at side and back scattering angles. As for ratio $-F_{12}(\theta)/F_{11}(\theta)$, small "milled loess" has smaller maximum values at near side scattering angles, while large "pristine loess" has relatively larger maximum values. Different from ratio $-F_{12}(\theta)/F_{11}(\theta)$, measured $F_{34}(\theta)/F_{11}(\theta)$ has larger maximum for small "milled loess" sample. Experimentally determined $F_{22}(\theta)/F_{11}(\theta)$ values of "milled loess" are larger than "pristine loess", especially at side and back scattering angles. It should be noted that discrepancies in measured $F_{22}(\theta)/F_{11}(\theta)$ cannot be directly used to indicate difference of particle irregularity, because optical calculations of Gaussian spheres showed that $F_{22}(\theta)/F_{11}(\theta)$ values are sensitive to not only particle irregularity but also to size distribution (Liu et al., 2015). As for ratios $F_{33}(\theta)/F_{11}(\theta)$ and $F_{44}(\theta)/F_{11}(\theta)$, the measurements for "milled loess" are larger than that for "pristine loess". In short, these discrepancies in scattering matrices between "pristine loess" and "milled loess" are inconsistent with that for all other kinds of dust with different size distributions in literatures.

In this study, several fundamental properties of loess dust samples were characterized for auxiliary analyses. As shown in Table 1, effective radii for "pristine loess" and "milled loess" are 49.40 μm and 2.35 μm, respectively. The real part of refractive index for "pristine loess" is 1.65 and that for "milled loess" is 1.70. Table 2 shows that the changes of chemical components are negligible. Therefore, it is reasonable to suspect that distinctions in angular distributions of measured scattering matrix elements for two loess samples may be mainly caused by different size distributions (effective radii differ by more than 20 times), while differences in other factors such as refractive index and micro structure have relatively small contributions in leading to different scattering matrices.

Literatures focused on optical modeling of irregular mineral dust were analyzed to find reasonable explanations for the differences in scattering matrix elements for "milled loess" and "pristine loess" samples. Numerical simulations of Gaussian spheres showed that as effective size parameter increases from 30 to 600, phase function $F_{11}$ as well as ratios $F_{33}/F_{11}$ and $F_{44}/F_{11}$ decrease, the maximum of ratio $F_{34}/F_{11}$ decreases and its negative branches at forward scattering and backscattering directions become small, the maximum of ratio $-F_{12}/F_{11}$ increases, and the ratio $F_{22}/F_{11}$ increases especially at backscattering angles (Liu et al., 2015). When Gaussian spheres become more non-spherical and irregular, phase function $F_{11}$ as well as ratios $-F_{12}/F_{11}$, $F_{22}/F_{11}$, $F_{33}/F_{11}$ and $F_{44}/F_{11}$ show different trends compared with the influences of increasing effective radius,

while the ratio $F_{34}/F_{11}$ show similar trend (Liu et al., 2015). Zubko et al. (2007) showed that as the surfaces of Gaussian particles become rougher, the ratio $-F_{12}/F_{11}$ tends to larger. Simulations of agglomerated debris particles showed that with the imaginary part of refractive index varies in the range 0-0.01, scattering matrix elements almost unchanged (Zubko et al., 2013). However, calculations of Gaussian particles conducted by Muinonen et al. (2007) showed that increase of refractive index (both real and imaginary part) leads to smaller $-F_{12}/F_{11}$ and $F_{22}/F_{11}$. In summary, different factors have different or similar effects on a certain matrix elements. The discrepancies in scattering matrices for "milled loess" and "pristine loess" can be mainly interpreted from the perspective of difference of effective radii, while differences in other factors such as refractive index and micro structure have relatively small contributions, and Gaussian spheres may be promising models for simulating scattering matrix for loess dust.

In this work, a relatively good case is presented to show the effect of size distribution of loess dust on scattering matrices because effective radii of "pristine loess" and "milled loess" differ by more than 20 times. The influence of loess particle size is roughly verified through qualitative analyses of simulation results of Gaussian sphere, which deepen the understanding of this effect. For more detailed explanations, quantitative analyses are still needed based on much more optical simulations of Gaussian spheres. However, besides size distribution, physical properties such as refractive index and micro structure also play important roles in determining scattering matrices of dust particles. When the difference in particle size distributions or effective radii is relative small, the influences of other factors may become dominant or un-ignorable. This may be the reason why the effect of size distribution on measured scattering matrices for olivine samples cannot be concluded clearly (Muñoz et al., 2000). And this may also be the reason why effective radii cannot be used to explain all the discrepancies in matrix elements for forsterite samples based on simulation results of Gaussian spheres (Volten et al., 2006b). Another reason may be that Gaussian spheres are not suitable models to reproduce scattering matrix for forsterite dust, as optical modelling of irregular mineral dust is still a challenging subject.

## 4.2 Synthetic Scattering Matrices

Laboratory measurements of scattering matrices only cover scattering angles from $5°$ to $175°$. In order to obtain scattering matrix over $0°$-$180°$, synthetic scattering matrices $F^{syn}$ are constructed by a combination of numerical simulation and extrapolation of experimental measurements (Dabrowska et al., 2015; Escobar-Cerezo et al., 2018).

Measured $F_{11}(\theta)$ values are normalized to 1 at $10°$, and these relative phase functions are the same for measured and synthetic scattering matrices for the same sample (Escobar-Cerezo et al., 2018):

$$\frac{F_{11}(\theta)}{F_{11}(10°)} = \frac{F_{11}^{syn}(\theta)}{F_{11}^{syn}(10°)} \tag{6}$$

where $F_{11}^{syn}(\theta)$ is the synthetic phase function that must fulfill the following normalized equation:

$$\frac{1}{2} \int_0^\pi d\theta \sin\theta F_{11}^{syn}(\theta) = 1 \tag{7}$$

SEM images for both loess samples show that most particles have relatively moderate aspect ratios. Therefore, Lorenz-Mie theory can be used to calculate forward peaks of synthetic phase functions at angles smaller than 5°. Because for particles who have moderate aspect ratios, forward peaks of synthetic phase functions mainly depend on size distributions and depend little on particle shapes (Liu et al., 2003). Refractive indices as well as size distributions for "pristine loess" and "milled loess" obtained from particle sizer are used in Lorenz-Mie calculations. For scattering angle 180°, multi-order polynomial extrapolation is used on the basis of experimentally determined relative phase functions. After then, the calculated forward peak of phase function as well as relative phase function after extrapolated are incorporated at 5° angle to construct synthetic phase function. Whether synthetic phase function satisfies Eq. (7) should be checked. Otherwise, increase or decrease measured relative phase function at 5° angle within measurement error, and repeat merging process and checking process until Eq. (7) is satisfied.

As for other matrix element ratios $F_{ij}(\theta)/F_{11}(\theta)$, a set of constraints at 0° and 180° scattering angles should be taken into consideration (Hovenier et al., 2014; Mishchenko and Hovenier, 1995):

$$\frac{F_{12}(0°)}{F_{11}(0°)} = \frac{F_{12}(180°)}{F_{11}(180°)} = \frac{F_{34}(0°)}{F_{11}(0°)} = \frac{F_{34}(180°)}{F_{11}(180°)} = 0 \tag{8}$$

$$\frac{F_{22}(0°)}{F_{11}(0°)} = \frac{F_{33}(180°)}{F_{11}(180°)} = 1 \tag{9}$$

$$\frac{F_{22}(180°)}{F_{11}(180°)} = -\frac{F_{33}(180°)}{F_{11}(180°)} \tag{10}$$

$$\frac{F_{44}(180°)}{F_{11}(180°)} = 1 - 2\frac{F_{22}(180°)}{F_{11}(180°)} \tag{11}$$

Synthetic values for ratio $F_{22}/F_{11}$ at 180° angle for "pristine loess" and "milled loess" are obtained by nine-order polynomial extrapolations. Then $F_{33}/F_{11}$ and $F_{44}/F_{11}$ at 180° are calculated according to Eqs. (10) and (11), respectively. In addition, right-hand (left-hand) derivative at 0° (180°) for each scattering matrix element must be 0 (Hovenier and Guirado, 2014). In Figure 5, synthetic matrices for "pristine loess" and "milled loess" are illustrated.

**Figure 5**

Using extrapolated value of $F_{22}/F_{11}$ at 180° angle, back-scattering depolarization ratio $\delta_L$ can be calculated, which is an essential parameter for aerosol lidar observations (Mishchenko et al., 2002):

$$\delta_L = \frac{F_{11}(180°) - F_{22}(180°)}{F_{11}(180°) + F_{22}(180°)} = \frac{1 - \frac{F_{22}(180°)}{F_{11}(180°)}}{1 + \frac{F_{22}(180°)}{F_{11}(180°)}} \tag{12}$$

Calculated back-scattering depolarization ratios for "pristine loess" and "milled loess" are 0.21 and 0.26, respectively, "milled loess" has larger value of $\delta_L$. Direct measurements of back-scattering depolarization ratios of Arizona Test Dust with different size distributions at both 355 and 532 nm wavelengths also showed that $\delta_L$ values for small particles are larger than that for large particles and this discrepancy is more pronounced at 532 nm (Miffre et al., 2016).

At last, the previously published average scattering matrix for loess, which consists of results for Hungary loess, milled Yangling loess and milled Luochuan loess (the latter two were sampled from CLP), was updated using new sample "pristine loess" from Luochuan, by averaging synthetic matrices for different loess samples. In other words, the differences between

345 average matrix before and after update are also the differences between "pristine loess" and the other three samples, and differences among these three samples can be referred to Liu et al. (2019). As shown in Figure 6, compared to other three samples, phase function for "pristine loess" has larger forward scattering peaks and smaller values at side and back scattering directions. "Pristine loess" has larger $-F_{12}(\theta)/F_{11}(\theta)$ values at near side scattering angles, has larger $F_{22}(\theta)/F_{11}(\theta)$ values at almost all scattering angles, and has smaller values of both $F_{33}(\theta)/F_{11}(\theta)$ and $F_{44}(\theta)/F_{11}(\theta)$ at backscattering directions, when

compared with the other three samples.

**Figure 6**

## 5 Conclusions

Asian dust contributes a lot to global atmospheric dust aerosol, and Chinese Loess Plateau (CLP) is a main origin of Asian dust. Loess dust aerosols originated from CLP are expected to affect the radiation balance potentially at both source

areas and downwind places far away from sources, because dust particles with different sizes can be transported over different distances. In this study, original loess sample was collected from Luochuan, which is centrally located at CLP. Subsequently, two loess samples with different size distributions were prepared for laboratory investigations. "Pristine loess" sample was used to represent loess dust that affect source regions only, and "milled loess" sample ball-milled from "pristine loess" was used to represent loess dust that can be transported over long distance. Light scattering matrices for both "pristine

loess" and "milled loess" samples at 532 nm wavelength were measured from 5° to 175° scattering angles. Besides particle size distribution, other basic properties were also characterized, such as chemical component, refractive index and microscopic appearance.

Even through experimentally determined angular behaviors of scattering matrix elements for "pristine loess" and "milled loess" are similar, there are still obvious discrepancies in matrix elements. More specifically, for small "milled

loess", relative phase function $F_{11}(\theta)/F_{11}(10°)$ as well as ratios $-F_{12}(\theta)/F_{11}(\theta)$ and $F_{22}(\theta)/F_{11}(\theta)$ are smaller than that for coarse "pristine loess", while ratios $F_{33}(\theta)/F_{11}(\theta)$, $F_{34}(\theta)/F_{11}(\theta)$ and $F_{44}(\theta)/F_{11}(\theta)$ are larger than that for coarse "pristine loess". These discrepancies are unique and different from that for other kinds of dust with distinct size distributions published in literatures. Qualitative analyses of optical simulations of various morphological model showed that the large difference in size distributions (effective radii differ by more than 20 times) caused by milling process plays a major role in

leading to discrepancies in scattering matrices for these two samples, while differences in factors such as refractive index and micro structure have relatively small and recessive contributions. And Gaussian sphere models may have good application prospect in optical modeling of loess dust, while more detailed quantitative verification using measured physical properties are still needed.

Synthetic scattering matrices for both "pristine loess" and "milled loess" were defined over 0°-180° scattering angle, and the previously presented average scattering matrix for loess was updated with new coarse "pristine loess" sample included. The phase function $F_{11}(\theta)$ in updated average matrix has larger forward scattering peaks and smaller values at side and backward scattering angles than that in previous average matrix. Compared to previous average matrix, updated average matrix has larger $-F_{12}(\theta)/F_{11}(\theta)$ at side scattering angles, has smaller $F_{33}(\theta)/F_{11}(\theta)$ and $F_{44}(\theta)/F_{11}(\theta)$ at backscattering angles. $F_{22}(\theta)/F_{11}(\theta)$ experiences the largest change before and after update, whose values are enlarged at almost all scattering angles.

In this study, scattering matrices for Chinese loess samples with large difference in their size distributions are investigated. Based on all the measurements, suitable shape distributions of spheroids can be obtained respectively, which are useful for the retrievals of airborne loess dust properties at both source and downwind areas in China or even East Asia. On the other hand, the updated average scattering matrix for loess are meaningful for the validation of exiting models and the development of more advanced morphological models suitable for loess dust, which are also useful to finally improve the retrieval accuracies of dust aerosol properties.

Fine loess dust sampled from Luochuan and Yangling, two regions of Chinese Loess Plateau, were investigated by Liu et al. (2019). Local variations of loess dust also have obvious effects on the measured scattering matrices. It should be noted that all these samples investigated may still cannot completely represent the loess in Chinese Loess Plateau and China, so one of the efforts in the future is to investigate more loess samples collected from more regions and with more size distributions, accordingly, the average scattering matrix for loess will be updated constantly. On the other hand, the validation of existing models and the development of more advanced models through reproducing measured scattering matrices using optical simulation results are also meaningful research directions.

**Data availability**

All the data involved in this study are available online at: https://github.com/liujia93/Scattering-matrix-for-loess-dust.

**Author contributions**

Jia Liu and Qixing Zhang designed the experiments; Jia Liu conducted the measurements; Yinuo Huo drew the layout diagram; all authors discussed the results; Jia Liu wrote the manuscript.

**Competing interests**

The authors declare that they have no conflict of interest.

## Acknowledgements

We are very grateful to Zidong Nie for loess dust sampling. We are also very grateful to Engineer Chao Li from HEFEI KE JING MATERIALS TECHNOLOGY CO., LTD. for milling dust particles. This work was financially supported by the National Natural Science Foundation of China (NSFC) (41675024 and U1733126), National Key Research and Development Program of China (2016YFC0800100 and 2017YFC0805100], and Fundamental Research Funds for Central Universities of China (WK2320000035).

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

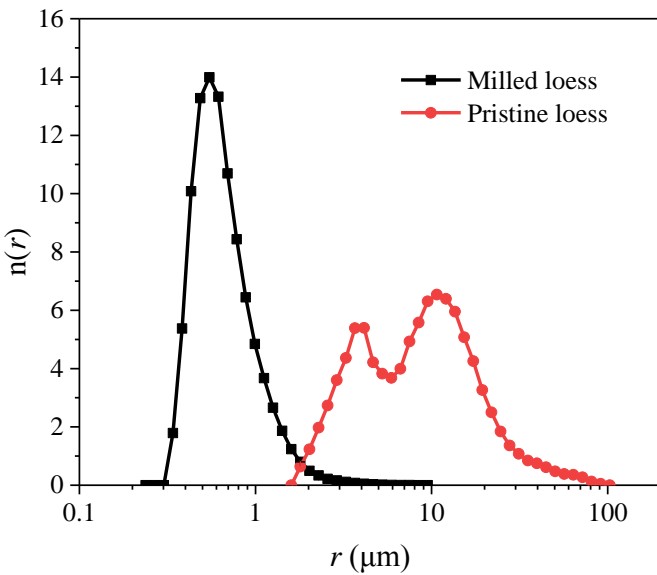

**Figure 1.** Normalized number size distributions n(*r*) of "pristine loess" and "milled loess". Radius *r* is plotted in logarithmic scale, and error bars are small and covered by symbols.


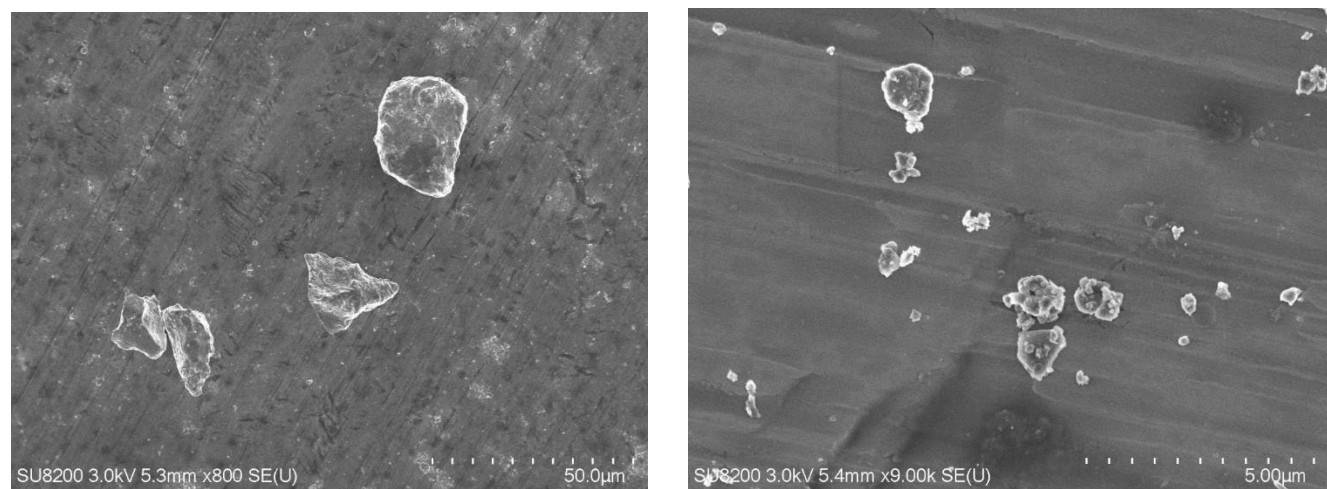

**Figure 2.** SEM images for "pristine loess" (left panel) and "milled loess" (right panel).

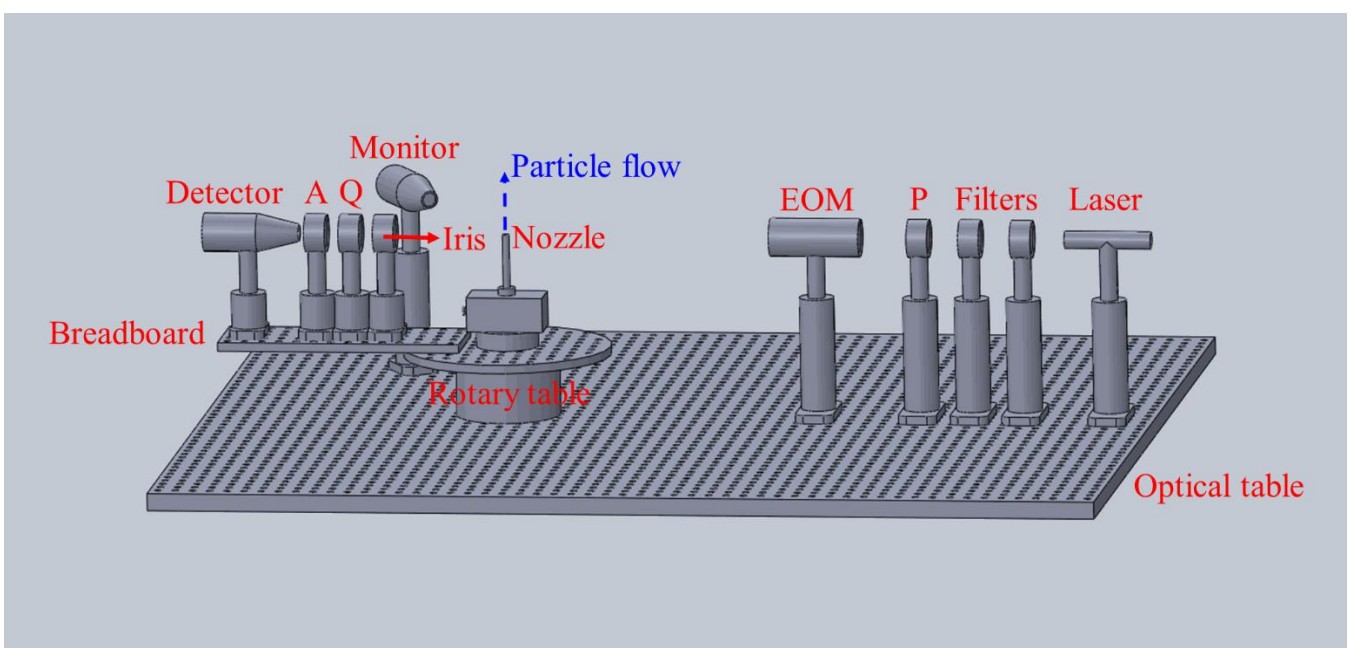

**Figure 3.** Layout diagram of the experimental apparatus after backscattering angle expended.

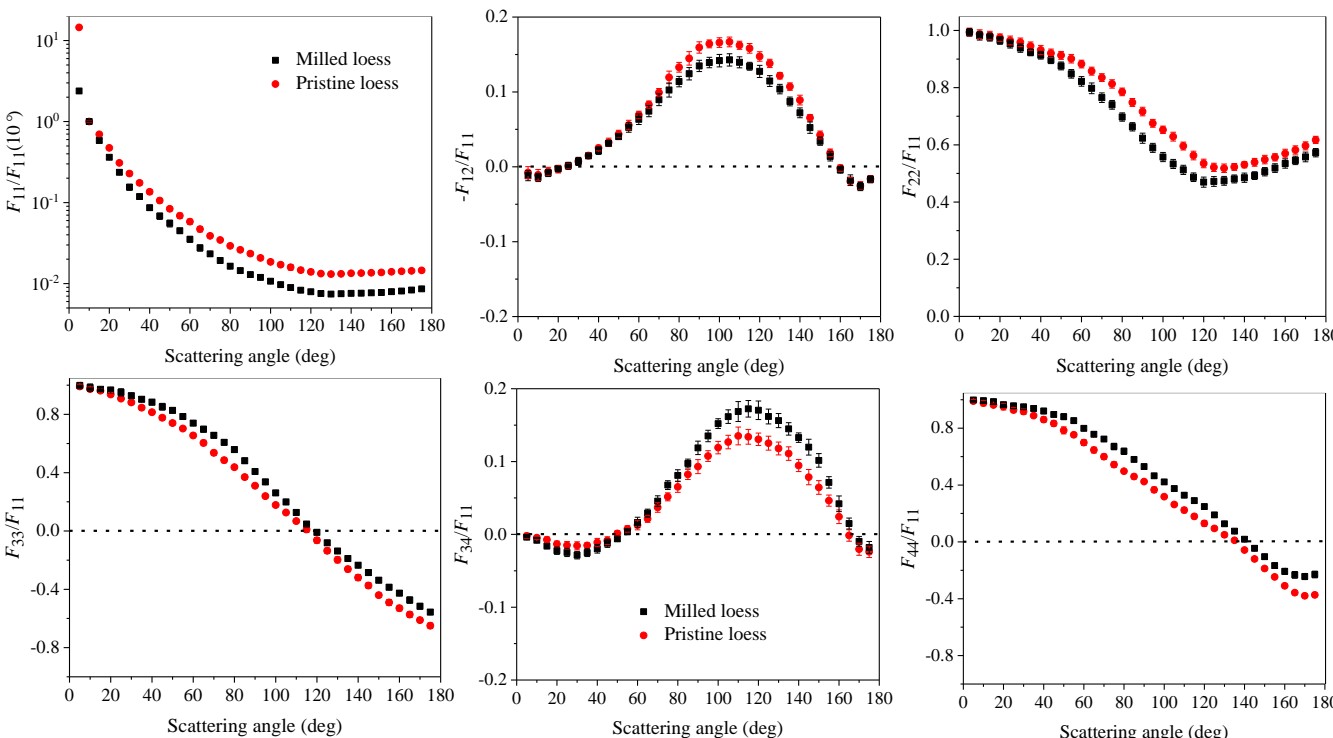

**Figure 4.** Measured non-zero scattering matrices for "pristine loess" and "milled loess". It should be noted that "milled loess" is the same sample as the "Luochuan loess" in Liu et al. (2019).


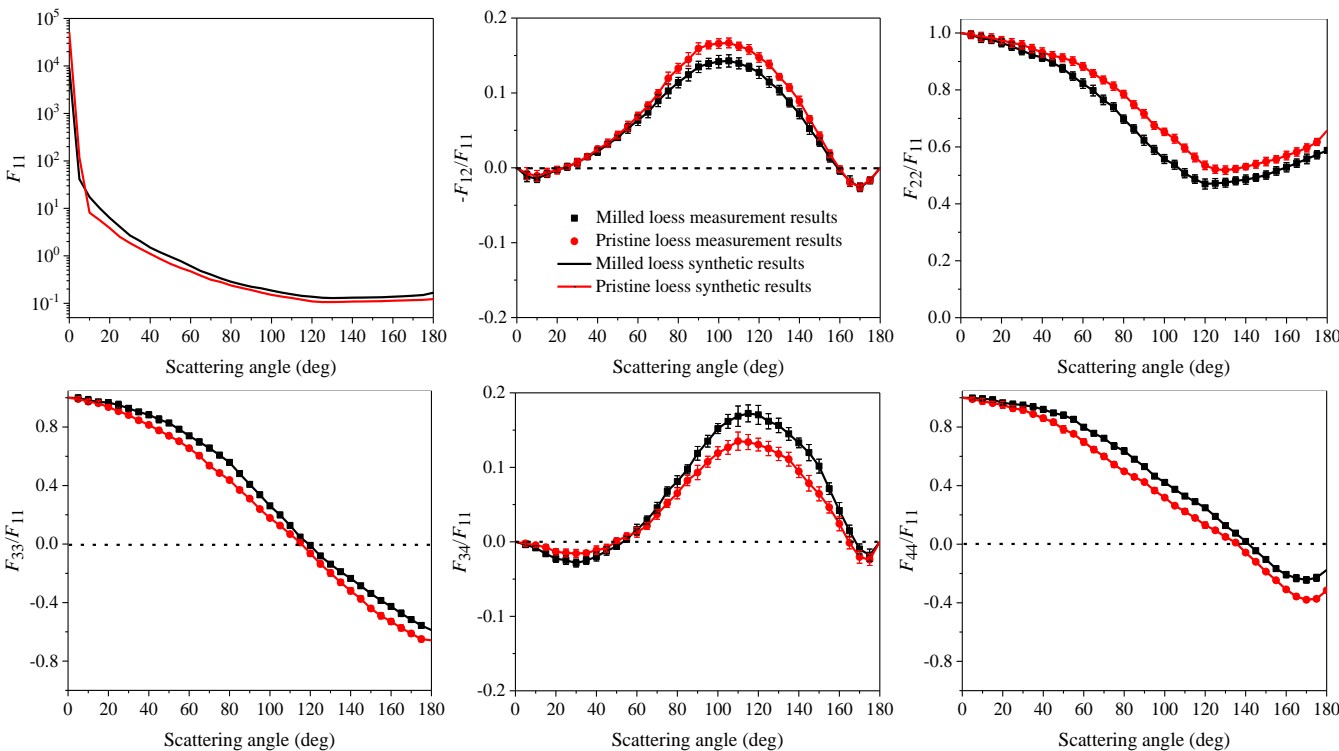

**Figure 5.** Synthetic scattering matrices for "milled loess" and "pristine loess". Lines are synthetic matrices and plots are measured values.

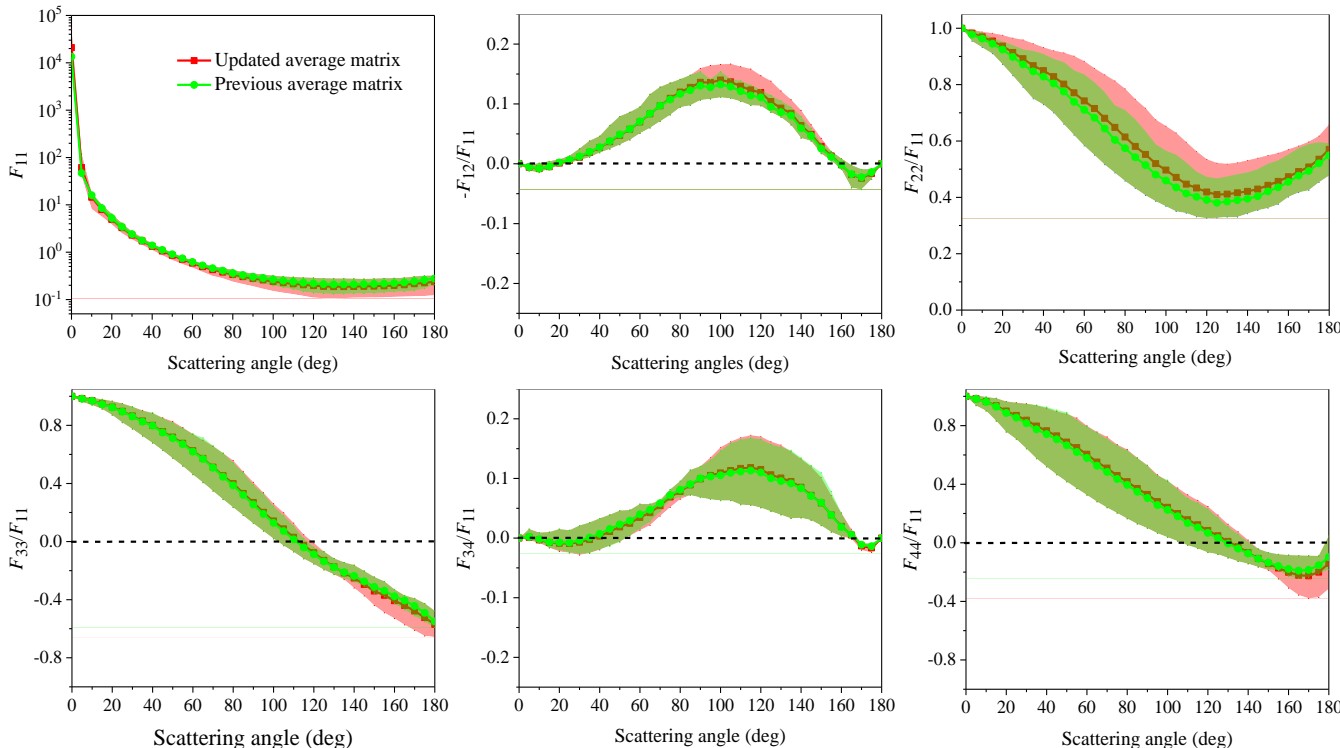


**Figure 6.** Previous average scattering matrix (green lines and solid circles) (Liu et al., 2019) and updated average scattering matrix (red lines and solid squares) for loess dust. Reddish and green shadows stand for the areas covered by results for different loess samples with or without "pristine loess" included, respectively.

**Table 1.** Size parameters and refractive indices of "pristine loess" and "milled loess".

| Samples | $r_{eff}$ (μm) | $\sigma_{eff}$ | $x_{eff}$ | $Re(m)$ | $Im(m)$ |
|---|---|---|---|---|---|
| Pristine loess | $49.40 \pm 1.98$ | $0.21 \pm 0.00$ | $583.2 \pm 23.7$ | 1.65 | 0 |
| Milled loess | $2.35 \pm 0.01$ | $0.64 \pm 0.00$ | $27.2 \pm 0.1$ | 1.70 | 0 |

**Table 2.** Chemical components of "pristine loess" and "milled loess" measured by XRF-1800.

| Components | Pristine loess (wt %) | Pristine loess error (wt %) | Milled loess (wt %) | Milled loess error (wt %) |
|---|---|---|---|---|
| $SiO_2$ | 63.8278 | 3.0237 | 66.2128 | 2.0900 |
| $Al_2O_3$ | 12.3091 | 0.3772 | 11.6487 | 0.2018 |
| $CaO$ | 9.2943 | 0.9455 | 7.8286 | 0.6450 |
| $Fe_2O_3$ | 5.5260 | 0.8817 | 5.6390 | 0.7411 |
| $K_2O$ | 3.3971 | 0.3004 | 3.3574 | 0.2358 |
| $MgO$ | 2.7536 | 0.4522 | 2.4843 | 0.2665 |
| $Na_2O$ | 1.2802 | 0.0243 | 1.3470 | 0.0214 |
| $TiO_2$ | 0.8017 | 0.0595 | 0.7939 | 0.0579 |
| $P_2O_5$ | 0.3340 | 0.0452 | 0.2549 | 0.0018 |
| $SO_3$ | 0.2370 | 0.1056 | 0.1687 | 0.0721 |
| $MnO$ | 0.1240 | 0.0294 | 0.1196 | 0.0120 |
| $ZrO_2$ | 0.0583 | 0.0104 | 0.0846 | 0.0122 |
| $SrO$ | 0.0348 | 0.0064 | 0.0299 | 0.0059 |
| $Rb_2O$ | 0.0177 | 0.0041 | 0.0174 | 0.0040 |
| $Co_2O_3$ | NT[*] | - | 0.0159 | 0.0049 |
| $Y_2O_3$ | NT[*] | - | 0.0061 | 0.0025 |

[*] NT: not detected.


**Table 3.** Combinations of orientation angles of optical axis of all the optical elements.

| Combination | $\gamma_P$ | $\gamma_{EOM}$ | $\gamma_Q$ | $\gamma_A$ | $DC(\theta)$ | $S(\theta)$ | $C(\theta)$ |
|---|---|---|---|---|---|---|---|
| 1 | 45° | 0° | - | - | $F_{11}(\theta)$ | $-F_{14}(\theta)$ | $F_{13}(\theta)$ |
| 2 | 45° | 0° | - | 0° | $F_{11}(\theta)+F_{21}(\theta)$ | $-F_{14}(\theta)-F_{24}(\theta)$ | $F_{13}(\theta)+F_{23}(\theta)$ |
| 3 | 45° | 0° | - | 45° | $F_{11}(\theta)+F_{31}(\theta)$ | $-F_{14}(\theta)-F_{34}(\theta)$ | $F_{13}(\theta)+F_{33}(\theta)$ |
| 4 | 45° | 0° | 0° | 45° | $F_{11}(\theta)+F_{41}(\theta)$ | $-F_{14}(\theta)-F_{44}(\theta)$ | $F_{13}(\theta)+F_{43}(\theta)$ |
| 5 | 90° | -45° | - | - | $F_{11}(\theta)$ | $F_{14}(\theta)$ | $-F_{12}(\theta)$ |
| 6 | 90° | -45° | - | 0 | $F_{11}(\theta)+F_{21}(\theta)$ | $F_{14}(\theta)+F_{24}(\theta)$ | $-F_{12}(\theta)-F_{22}(\theta)$ |
| 7 | 90° | -45° | - | 45° | $F_{11}(\theta)+F_{31}(\theta)$ | $F_{14}(\theta)+F_{34}(\theta)$ | $-F_{12}(\theta)-F_{32}(\theta)$ |
| 8 | 90° | -45° | 0° | 45° | $F_{11}(\theta)+F_{41}(\theta)$ | $F_{14}(\theta)+F_{44}(\theta)$ | $-F_{12}(\theta)-F_{42}(\theta)$ |