# Peer review of "An experimental study on light scattering matrices for Chinese loess dust with different particle size distributions"

_Atmospheric Measurement Techniques, 2019_

## Referee Comment (RC1) · Anonymous Referee #1 · 10 Aug 2019

Experimental studies like this one are still rare and should be encouraged. This is a useful paper and can be published largely as is. I would only suggest to expand the motivation for this study in the introduction by pointing out that satellite retrievals of dust-aerosol characteristics such as, e.g., the optical thickness are strongly affected by particle nonsphericity (e.g., [1]), and so reliable knowledge of the phase function (or, more generally, the scattering matrix) for real dust aerosols is essential.

[1] Mishchenko, M. I., I. V. Geogdzhayev, L. Liu, J. A. Ogren, A. A. Lacis, W. B. Rossow, J. W. Hovenier, H. Volten, and O. Munoz, 2003: Aerosol retrievals from AVHRR radiances: effects of particle nonsphericity and absorption and an updated long-term

global climatology of aerosol properties. J. Quant. Spectrosc. Radiat. Transfer 79/80, 953-972.

---

## Author Comment (AC1) · 8 Oct 2019

The authors really appreciate the valuable comments and constructive suggestions from the reviewer. The suggestions and comments of reviewer are listed in black font, and responses are highlighted in blue. The changes made in the revised manuscript are marked in red font.

**Comments from reviewer 1:**

Experimental studies like this one are still rare and should be encouraged. This is a useful paper and can be published largely as is. I would only suggest to expand the motivation for this study in the introduction by pointing out that satellite retrievals of dust-aerosol characteristics such as, e.g., the optical thickness are strongly affected by particle nonsphericity(e.g., [1]), and so reliable knowledge of the phase function (or, more generally, the scattering matrix) for real dust aerosols is essential.

[1] Mishchenko, M. I., I. V. Geogdzhayev, L. Liu, J. A. Ogren, A. A. Lacis, W. B. Rossow, J. W. Hovenier, H. Volten, and O. Munoz, 2003: Aerosol retrievals from AVHRR radiances: effects of particle nonsphericity and absorption and an updated long-term global climatology of aerosol proper ties. J. Quant. Spectrosc. Radiat. Transfer 79/80, 953-972.

Response:

Thank you for reviewing our manuscript and the constructive comments.

We have expanded the motivation of our study in the revised manuscript as follows:

"It is well known that dust particles have distinct non-spherical shapes, and thus retrievals of dust aerosol properties, like optical thickness, based on Lorenz-Mie computation will lead to large errors (Mishchenko et al., 2003). Dubovik et al. (2006) employed a mixture of spheroids with different axial ratios as well as spheres to reproduce laboratory measured angular scattering patterns of dust aerosols published by Volten et al. (2001), and the best fitted shape distribution of spheroids was obtained and recommended. Subsequent studies on the retrievals of aerosol properties from space-based (Dubovik et al., 2011), airborne (Espinosa et al., 2019) and ground-based (Titos et al., 2019) remote sensing observations were all based on this shape distribution. However, the assumption of a same shape distribution of spheroids is somewhat too arbitrary (Li et al., 2019) and may not be suitable for reproducing optical properties of loess dust particles with different size distributions. Therefore, laboratory measurements of optical properties, especially angular scattering patterns, as well as other basic physical features, like size distribution, refractive index and micro structure, of loess dust with different sizes are essential and beneficial to the development of more appropriate and precise optical models for this kind of dust. This will further help to retrieval more accurate aerosol properties at both dust 
[revised manuscript text omitted]

---

## Referee Comment (RC2) · Anonymous Referee #3 · 11 Dec 2019

General Comments:

This study presents an original measurement of dust samples and therefore fulfils the criterion of novelty. As it additionally presents a combination of techniques that can be seen as a new method, it fits the scope of AMT. While the paper still needs some improvement, the methods are ultimately fine. There are some weaknesses as to the significance of the work and the conclusions that are drawn, but these can probably be targeted by clearly stating the limits and some more explanation. The language is mostly fluent and precise. However, there are a still lot of mistakes. These can be fixed easily. The manuscript would benefit from having a native speaker or professional En-

glish proofreader go over it in detail. If the comments can be addressed appropriately, I recommend publication.

- The complete analysis is based on one single sample. This is a major weakness of the study. Yet as this is unlikely to be corrected retroactively, I suggest to discuss this fact thoroughly and state the limitations of the study. How representative is this sample of the Chinese Loess Plateau? There must be local variations, and the fact that it was sampled from the middle (page 3, line 93) does not make it representative per se. The limitation of drawing and measuring just one single sample have to be stated clearly.

- The original sample is milled to produce smaller particles that may be transported further. Why is it milled to the given size, not larger and not smaller? The study shows significant change of dust properties with size, and the milled loess seems to be just an arbitrary size.

- It is not clear enough what the conclusion of the study is. Scattering matrices are reported, but what do they ultimately tell us about the Chinese loess dust?

Specific comments:

- page 2, line 34-35: Please rephrase "It is common knowledge that ...". Literature that proves the statement is provided in the next paragraph, so there is no need to rely on "common knowledge".

- page 2, line 38 it should be "...CLP is expected to have important influence" instead of "...CLP will have important influence", as the statement is not proven.

- page 2 and 3, literature values for scattering matrix: Please elaborate on what the scattering matrix tells us, which properties do $F_{ij}$ and their quotients describe? Explain either here or in section 3.1.

- page 4, line 120: SEM "images", instead of "photographs", as this is an imaging technique detecting electrons, not photons.

- Table 2 and paragraph 1 on page 5: Are the differences in the sample composition significant? What are the errors on this analysis?

- page 6, section 3.2: Add some more detail of how the analysis was done. How many measurement iterations were performed, how are the final results derived from these?

- page 7, section 4.1: Similar as in the introduction, it should be discussed what the physical meaning of the results are. This is partly attempted in line 199, but should be done more thoroughly.

- page 8, lines 216-217 Please add the units of the parameters.

- page 8, lines 235-240 The description is rather vague, please make it clear what your actual finding is.

- page 10, line 302: As in page 2, line 38: Rather write "is expected to affect" or similar instead of "will affect".

- page 11, paragraph 2: Please make it more clear what the scattering matrices tells us. This section is now more a summary than a conclusion.

- page 11, line 323: Data availability: You uploaded the data, which is great, this should be linked here.

- Table 1: Add units.

- Abstract and Conclusions: Please add: What do the results of that study actually tell us about light scattering by Chinese loess in one sentence?

———————————————

---

## Referee Comment (RC3) · Anonymous Referee #4 · 17 Dec 2019

The study presents light scattering measurements of Chinese loess dust. The authors have measured the scattering matrix elements of a single loess sample from the Chinese loess plateau once untreated (pristine loess) and once milled (milled loess) and performed some complementary measurements too. I find the topic very interesting and useful. However, I have some doubts about the paper being published in its current form.

First of all I have to question if the choice of the journal is adequate for the performed study. I might be wrong about this and if this is the case, then please just ignore this comment. However, this journal is called Atmospheric Measurement Techniques,

and on its homepage it is stated that: "The main subject areas comprise the development, intercomparison, and validation of measurement instruments and techniques of data processing and information retrieval for gases, aerosols, and clouds." This paper presents none of them. It shows some laboratory measurements with atmospheric relevance. It does not show a new measurement technique nor a newly developed instrument neither any instrument intercomparison. The only technical part of the paper is the one page section of 3.2 where the measurement apparatus is shortly introduced.

My other main concern is: if the manuscript contains strong enough scientific material to be published in AMT. The scattering matrix element measurements of the two differently treated loess sample come from 6 single measurements, and the manuscript is based completely on this. It would considerably improve the manuscript if more measurements were included. To give you some ideas: include measurements and a comparison of different kind of loess samples collected either on the Chinese Loess Plateau at other places or get loess samples from outside of China. Another idea could be to include some other types of mineral dust and make a comparison. I know well, that it is not always possible to perform more measurements additionally. The manuscript could be improved with much thorougher discussion about comparing existing literature data with your dataset as well, or perform some numerical simulations based on the measured size distribution and shape (e.g. Mie theory and a theory for non-spherical particles) and discuss the results.

You could also improve the paper by stating clearly what your main message is for the reader. You just present the scattering matrix elements but do not draw any further conclusions. How is Chinese loess scattering treated in radiative transfer models? Will there be a big difference if these models are updated with your results? How representative is your single loess sample?

You probably cannot implement all of my main suggestions to improve the manuscript, and it is not necessary either. I just wanted to show you some possible options how it could be done. The data and the work you do is valuable but I only can recommend

the manuscript's publication if it is significantly improved.

Other Comments:

1. I suggest a careful English language editing of the manuscript.

2. Page 4, Lines 99-111: Even after a longer search I could not find details about the SALD-2300 instrument and how it exactly measures the particle size distribution and refractive index. Please add details how it exactly works. What I think it does is measuring the light scattering at many angles and trying to reproduce the measurement with a guessed number size distribution and a refractive index using theoretically calculated scattering values. Does it use the Mie theory (which is valid for spherical particles only)? Or how can it calculate the scattering for particles with unknown shape? How does it influence your derived number size distribution and refractive index? What is the uncertainty of this measurement method for non-spherical particles? Please add a discussion on this. Are you sure that the refractive index difference between 1.65+0i for "pristine loess" and 1.70+0i for "milled loess" is real?

3. Page 4, Lines 110-111: "larger particles have relatively larger real part of refractive index": if I understood correctly your method of producing the milled loess sample, it contains exactly the same material (your chemical analysis verifies it) as the pristine loess and therefore one would expect the two samples having the same refractive index. Are you sure, again, that your result is real and are not only a measurement artifact/uncertainty? Or do you think that the milling caused some strange structural changes in the loess sample which homogenized or inhomogenized how the chemical components are distributed within a single particle and/or between the particles?

4. Page 6, Section 3.2: I assume that this is not the first paper which uses this experimental apparatus. Please add a reference to the paper where a more detailed description of your instrument is available. If there is no such paper, please add a more detailed description.

5. Page 6, Lines 59-62: Since your main results are the measured matrix elements, probably it would be worth explaining exactly from which polarization states which matrix elements were derived and how, and not only referencing a paper for it.

6. Page 7, Line 193: "all six non-zero matrix elements are limited to narrow regions, respectively" I don't understand what you mean here. What narrow regions? Angle range? Y-value range? Or do you mean that your error bars are small? Please clarify!

7. Page 7, Lines 199-201: Please comment on the angular behavior of F_22. Next to it: it looks like that the milled loess sample deviates more from unity than the pristine loess sample. Does this suggest that the milled loess has a more irregular shape than pristine loess?

8. Page 7, Lines 206-207: The sentence is very confusingly phrased, please rephrase it. I am not sure if I understood what you wanted to tell the reader but I don't see any significant difference between the 5° relative phase functions.

9. Page 7, Lines 206-208: From F11 it looks like that milled loess has a higher forward to backward scattering ratio than pristine loess. I would expect exactly the other way around because the pristine loess sample contains much larger particles and larger particles usually have a much higher forward scattering compared to the backward scattering value. Please comment on it.

10. Page 8, Lines 119-223: Is there no way to produce samples containing smaller particles than the original without changing their form? Just by sieving the sample (the size distribution of the pristine loess seems to me broad enough)? Would that not work? If it would, then measuring such samples could save you from speculating about, if the measured differences are due to the different size or shape. It would be also very nice to have more samples with different sizes and not only two. You show that the particle size differs much more than the shape between the two samples, and your speculations might be true as well. However, how can you be sure that every component of the scattering matrix is comparably sensitive to the changes in size and

shape? Let's assume, that one matrix component is 1000 times more sensitive to the changes in the particle shape than to the changes in the size? Please provide some proof that such a case is not to be expected, and then your argumentation becomes valid.

11. Page 8, Lines 224-240: It would considerably strengthen the manuscript if numerical calculations based on your measured size distribution and particle shape were added and not only the existing literature was analyzed. If that is not possible, you should show how the size and shape of your samples compare to the size and shape of the particle in the referenced papers. Irregular dust does not necessary mean comparable size distribution and/or particle shape.

12. Page 9, Section 4.2: During calculating the synthetic scattering matrices you follow the works of Dabrowska et al., 2015 and Escobar-Cerezo et al., 2018. They used the very same measurement technique, had only different kind of samples (Lunar and Martian dust). You clearly follow their work, by extrapolating the measurements to the angles you could not measure as well. The extrapolation in the forward direction is based on the Mie theory and is performed for a narrow angle range of 0-3° or 0-5° (in your case). This is for me a justified assumption. However, in the backward region, the extrapolation is based on a polynomic fit and not on any kind of scattering theory. In this case, I can believe that it works well for the very narrow 177-180° angle range in the works of Dabrowska et al., 2015 and Escobar-Cerezo et al., 2018. But you applied it for a much broader angle range of 160-180°, and here I really need some solid proof of this method being justified. The later calculated back-scattering depolarization ratio values cannot be accepted either before your extrapolation is not verified.

Technical Comments: I did not do any language/technical correction because the manuscript needs a bigger revision.
* * *

---

## Referee Comment (RC4) · Anonymous Referee #5 · 17 Dec 2019

General Comments:

In Liu et al. the paper focuses on describing the scattering function of a sample that was collected from the Chinese Loess Platea and subsequently milled to change the physical properties of the particles. The major conclusion gleaned by the authors in the article is that the size of the particles affects the scattering properties. The paper does describe well the need for the research being performed on complex systems, but systematic experiments need to be performed to start to tease out some of that information instead of broad statements about size since that is what they were trying to control. The authors mention that the size distribution is the major factor, but refractive

index and micro structure are not ignorable (line 237-240) and then seem to discount that the shape refractive index have little effect (line 318). They additionally mention that the Refractive index is different (Table 1), but do not seem to try and account for the difference using any kind of modeling to show that it is primarily size. Or identify as to why these are different for the same material.

This paper does not show significant new data or a new approach to understanding the optical properties of aerosol particles that had not been published previously by the group. The technique has been described by the authors at least twice previously in prior publications and one of the 2 sample sets is already published elsewhere (Liu et al. 2019 and 2018). The paper itself needs to be edited further and reorganized as there are multiple sections that are very similar but spread out through the paper. This paper appears to be more of an addendum to the Liu et al 2019 article than a stand-alone article. Based on these above points, I would be hesitant to recommend this paper for publication as is since there is little information that is novel and there are some unsubstantiated claims throughout.

Specific Comments:

Line 26: Please specify what this % is, from written it appears to be total aerosol loading worldwide.

Line 36: Please specify what 'r' refers to specifically.

Line 42: Remove "Without a doubt"

Line 55: 'Furthermore. . .. scattering matrices' This sentence is not completely coherent and needs to be rewritten.

Line:99-100: What was the injection type for the laser particle sizer? Were they injected in solution or dry?

Line 101: Size comparison can be difficult between the two samples due the fact that the original dust sample has a bimodal distribution. This distribution itself will lead to

very different scattering properties, whereas the milled sample is a more uniform size. What is the cause of the bimodal shape? Could this be due to a heterogeneity of mineral types being different sizes and having large differences in scattering properties that are then not comparable to the milled sample?

Line 103: It is stated that the majority of the particles are larger than 5 microns, but there is a peak at 3 and 10 microns. Please reword this section because you use a cutoff of 5 microns earlier for local vs. long range transport.

Line 105: Please define the peaks more clearly for both samples, with a peak maximum and additional parameters to describe the spread.

Line 110/Table 1: Why is there a difference in the refractive index if they are still the same material? Please provide the error associated with the measurements and propagate through the rest of the calculations.

Line 120: how are the samples for SEM prepared? Are they impacted on the surface or collected some other way?

Line 129: What is the detection limit of this instrument? You quote down to 0.0001 wt% in Table 2. This is mainly of interest since I do not know the limits of XRF.

Table 2: add an additional column with the difference between the pristine and milled samples. Also include that the characterization was performed by XRF in the caption.

Line 124: The aggregated particles are all on the large size of the size distribution, would this affect the scattering properties greatly or are they artefacts from particle collection for SEM analysis?

Figure 2: The SEM image for the pristine loess only shows particles in the 10s of microns, it is not a representative image of what the particles actually would look like since the peaks are at $\sim$3 and 10 microns. Additionally, the image for the 'milled loess' is the same as previously published in the prior manuscript. Please provide representative and comparative SEM images.

Line 126: What size ZrO2 ball were used and were they milled wet or dry?

Line 153-160: I like that the detectors are defined differently, but it would be better to have a different description that 'monitor' and 'detector' as they are both the same pmt detectors just with different functions.

Figure 3: This does not seem necessary as the technique has been described twice previously.

Lines: 215-223: this paragraph is in an odd place as it references past tables and figures.

218: "loess dust become more irregular after milling process" How is this defined? If you are saying that they become more irregular, then you will need to actually do analysis of the particles themselves to show the change in the shape parameters. Based on the images seen, this statement cannot be made.

Line 241-253: This paragraph could be combined with the conclusion, it is very repetitive.

Figure 4/5: Could these be combined? You could have the synthetic scattering matrix as a different color and a line. It took me a while to see what the difference was between the 2 figures.

Figure 6: Could you specify all the samples that were used in this figure? Either here or in the text.

318-319: "other factors..." this is misleading, since there was no discussion on how the difference in RI affected the sample and no experiments were performed to single these factors out from the size effect. This is also in contrast to earlier where it is stated in line 239-240 "while other factors are also not ignorable"
* * *

---

## Referee Comment (RC5) · Anonymous Referee #6 · 20 Dec 2019

General Comments: The manuscript by Liu et al. presents results from the light scattering matrices for the samples collected from Chinese Loess Plateau. Auxiliary analyses including particle size distribution, refractive index, chemical component, and microscopic appearance etc. were also done. Based on their results, the authors conclude that the size distribution play a major role in leading to different matrices. In general, the method developed by the authors is novel and fits the slope of the journal. However, some modifications are necessary before it can be considered for publication. One major comment is that the authors did not discuss the atmospheric implications of this novel method. The authors mentioned that the average scattering matrix changed due to the updated sample "pristine loess" compared to previous studies (Fig. 6), this

is very interesting, but how meaningful this is to the atmospheric aerosols study? How accurate will it be if we use this new average scattering matrix in future studies? Also, it is necessary for the authors to ask a native English speaker to review the article.

Specific comments:

Pg4, line 100: Figure 1: What is the meaning of r in y-axis? Please explain in the figure caption.

Pg6, Fig. 3 is not so clear. Please draw a schematic of the experimental setup.

Pg6, line 155: How do you inject the dust aerosols into the setup? Please clarify.

Pg8, line 222: The last sentence "while other..." is not clear.

Pg11, line 313-314: The authors should indicate what are these small "milled loess" compared to.

Pg10-11, In the conclusion section, the authors should explicitly explain what is "novel" in the new average scattering matrix compared to their previous study, and the significance of this study.

———————————————————

---

## Author Comment (AC2) · 14 Jun 2020

**Responses to the comments of reviewer 1**

The authors really appreciate the valuable comments and constructive suggestions from the reviewer. The suggestions and comments of reviewer are listed in black font, and responses are highlighted in blue. The changes made in the revised manuscript are marked in red font.

**Comments from reviewer 1:**

Experimental studies like this one are still rare and should be encouraged. This is a useful paper and can be published largely as is. I would only suggest to expand the motivation for this study in the introduction by pointing out that satellite retrievals of dust-aerosol characteristics such as, e.g., the optical thickness are strongly affected by particle nonsphericity (e.g., [1]), and so reliable knowledge of the phase function (or, more generally, the scattering matrix) for real dust aerosols is essential.
[1] Mishchenko, M. I., I. V. Geogdzhayev, L. Liu, J. A. Ogren, A. A. Lacis, W. B. Rossow, J. W. Hovenier, H. Volten, and O. Munoz, 2003: Aerosol retrievals from AVHRR radiances: effects of particle nonsphericity and absorption and an updated long-term global climatology of aerosol proper ties. J. Quant. Spectrosc. Radiat. Transfer 79/80, 953-972.

Response:

   Thank you very much for reviewing our manuscript and the constructive comments. We have expanded the motivation of our study in the Introduction in the revised manuscript:

[revised manuscript text omitted]

---

## Author Comment (AC3) · 14 Jun 2020

Responses to the comments of reviewer 3

The authors really appreciate the valuable comments and constructive suggestions from the reviewer. The suggestions and comments of reviewer are listed in black font, and responses are highlighted in blue. The changes made in the revised manuscript are marked in red font.

**Comments from reviewer 3:**

General Comments:
This study presents an original measurement of dust samples and therefore fulfils the criterion of novelty. As it additionally presents a combination of techniques that can be seen as a new method, it fits the scope of AMT. While the paper still needs some improvement, the methods are ultimately fine. There are some weaknesses as to the significance of the work and the conclusions that are drawn, but these can probably be targeted by clearly stating the limits and some more explanation. The language is mostly fluent and precise. However, there are a still lot of mistakes. These can be fixed easily. The manuscript would benefit from having a native speaker or professional English proofreader go over it in detail. If the comments can be addressed appropriately, I recommend publication.

Response:
    Thanks a lot for reviewing our manuscript and all these constructive comments. We have responded your comments point by point and modified related descriptions in the revised manuscript. In addition, we have tried our best to correct languages mistakes by checking our manuscript repeatedly and inviting native speakers to review it. We hope that you will reconsider our manuscript.

- The complete analysis is based on one single sample. This is a major weakness of the study. Yet as this is unlikely to be corrected retroactively, I suggest to discuss this fact thoroughly and state the limitations of the study. How representative is this sample of the Chinese Loess Plateau? There must be local variations, and the fact that it was sampled from the middle (page 3, line 93) does not make it representative per se. The limitation of drawing and measuring just one single sample have to be stated clearly.

Response:
    Thank you for the valuable comments.
    As mentioned in manuscript, our original loess sample was collected from Luochuan Loess National Geological Park, which is the only national park for loess landform in China. So we think the sample represents Chinese loess to some extent, but it still cannot represent all loess distributed in China, even all loess in Chinese Loess Plateau.
    In our another work (Liu et al., 2019), we investigated fine loess particles sampled from Luochuan and Yangling, which located at the southern edge of Chinese Loess Plateau. Results showed that discrepancies in their scattering matrices are also obvious and even larger than that for Luochuan samples with different sizes, which means the

effect of local variations of loess on scattering matrices are also significant. When we tried to explain the discrepancies for loess sampled from different sites based on analyses of numerical simulations, we found it is hard to summarize which physical property (size distribution, micro structure, and refractive index) plays a major role, because there are no significant differences in these properties in our opinion. Because difference in size distributions have significant effects on scattering matrices for dust, and particles with different sizes are relatively easy to obtain compared to other properties. Therefore, we investigated scattering matrices for loess dust with large difference in their sizes distributions in this study to further explore explanations of discrepancies in scattering matrices based on analyses of numerical simulations, which is significant from the perspective of particle transportation.

In short, local variations of loess are also important and worthy of extended investigations, but this is slightly different from the motivation of this work. So we would like to conduct this extended research in our future work, representative samples from more regions of Chinese Loess Plateau (even China) with various size distributions will be investigated, and the average scattering matrix will be updated constantly.

We have modified related descriptions in Section 2 and added necessary discussions in Conclusions in the revised manuscript:

"Original loess dust sample was collected from Loess National Geological Park (35.76 °N, 109.42 °E) at Luochuan, which is lying on "loess zone" and also at the center of CLP. Since this park is the only national geological park in China which has typical loess geomorphology, it can be considered that the sample collected represents Chinese loess to a certain extent."

"Fine loess dust sampled from Luochuan and Yangling, two regions of Chinese Loess Plateau, were investigated by Liu et al. (2019). Local variations of loess dust also have obvious effects on the measured scattering matrices. It should be noted that all these samples investigated may still cannot completely represent the loess in Chinese Loess Plateau and China, so one of the efforts in the future is to investigate more loess samples collected from more regions and with more size distributions, accordingly, the average scattering matrix for loess will be updated constantly."

- The original sample is milled to produce smaller particles that may be transported further. Why is it milled to the given size, not larger and not smaller? The study shows significant change of dust properties with size, and the milled loess seems to be just an arbitrary size.

Response:

Thanks a lot for your comments. We acknowledge that the milled fine sample actually has an arbitrary size distribution. Because it is almost impossible to obtain loess samples with preset sizes and size distributions by ball milling. Although particle size distributions of samples can be roughly changed by adjusting milling time, the particle sizes of finally obtained samples are still arbitrary in nature. Even through the size distribution of milled sample is kind of arbitrary, since this sample satisfies criterion for particle long range transportation, so the investigation of this sample still useful for developing optical models of fine loess dust.

- It is not clear enough what the conclusion of the study is. Scattering matrices are reported, but what do they ultimately tell us about the Chinese loess dust?

Response:

Thank you for the constructive comments.

In this study, we paid more attention to present the discrepancies in scattering matrices for Chinese loess dust with different size distribution and tried to find explanations for these discrepancies based on analyses of optical simulation results. The results and conclusions include the following three aspects: (1) there are obvious discrepancies in measured scattering matrices for Chinese loess dust with different size distributions, and these discrepancies are different from that for other kinds of mineral dust with various size distributions. (2) Qualitative analyses of numerical simulation results in literatures showed that the large difference in size distributions (effective radii differ by more than 20 times) plays a major role in leading to these discrepancies in scattering matrices. And Gaussian spheres may be promising models for simulating scattering matrix for Chinese loess dust, but more detailed quantitative verifications using measured size distributions and refractive indices are still needed. (3) The previously published average scattering matrix for loess dust was updated using measurements of new coarse loess sample, which is meaningful for validating existing models and developing more advanced models suitable for optical simulations of loess dust, and finally helps to retrieve dust aerosol properties with higher accuracy over both source and downwind areas.

We have modified and added related descriptions in Abstract and Conclusions in the revised manuscript to make the conclusions of our study more clear:

"Experimental results showed that there are obvious discrepancies in angular behaviours of matrix elements for "pristine loess" and "milled loess", and these discrepancies are different from that for other kinds of dust with distinct size distributions. Given that the effective radii of these two loess samples differ by more than 20 times, it is reasonable to conclude that the difference in size distributions plays a major role in leading to different matrices, while differences in refractive index and micro structure have relatively small contributions. Qualitative analyses of numerical simulation results of irregular particles also validate this conclusion. Gaussian spheres may be promising morphological models for simulating scattering matrix of loess but need further quantitative verification. At last, synthetic scattering matrices for both "pristine loess" and "milled loess" were constructed over 0°-180°, and the previous average scattering matrix for loess dust was updated. This study presents measurement results of Chinese loess dust and updated average scattering matrix for loess, which are useful for validating existing models and developing more advanced models for optical simulations of loess dust and finally help to improve retrieval accuracy of dust aerosol properties over both source and downwind areas."

"These discrepancies are unique and different from that for other kinds of dust with distinct size distributions published in literatures. Qualitative analyses of optical simulations of various morphological model showed that the large difference in size distributions (effective radii differ by more than 20 times) caused by milling process

plays a major role in leading to discrepancies in scattering matrices for these two samples, while differences in factors such as refractive index and micro structure have relatively small and recessive contributions. And Gaussian sphere models may have good application prospect in optical modeling of loess dust, while more detailed quantitative verification using measured physical properties are still needed."

"Synthetic scattering matrices for both "pristine loess" and "milled loess" were defined over 0°-180° scattering angle, and the previously presented average scattering matrix for loess was updated with new coarse "pristine loess" sample included. The phase function $F_{11}(\theta)$ in updated average matrix has larger forward scattering peaks and smaller values at side and backward scattering angles than that in previous average matrix. Compared to previous average matrix, updated average matrix has larger -$F_{12}(\theta)/F_{11}(\theta)$ at side scattering angles, has smaller $F_{33}(\theta)/F_{11}(\theta)$ and $F_{44}(\theta)/F_{11}(\theta)$ at backscattering angles. $F_{22}(\theta)/F_{11}(\theta)$ experiences the largest change before and after update, whose values are enlarged at almost all scattering angles."

"In this study, scattering matrices for Chinese loess samples with large difference in their size distributions are investigated. Based on all the measurements, suitable shape distributions of spheroids can be obtained respectively, which are useful for the retrievals of airborne loess dust properties at both source and downwind areas in China or even East Asia. On the other hand, the updated average scattering matrix for loess are meaningful for the validation of exiting models and the development of more advanced morphological models suitable for loess dust, which are also useful to finally improve the retrieval accuracies of dust aerosol properties."

Specific comments:
- page 2, line 34-35: Please rephrase "It is common knowledge that ...". Literature that proves the statement is provided in the next paragraph, so there is no need to rely on "common knowledge".

Response:
   Thank you for the suggestions. We have modified the related descriptions in the revised manuscript:
   "Dust particles with different sizes can be transported over different distances, more specifically, dust particles with a size range of r > 5 μm exist in source areas only, while particles with a size range of 0.1 < r < 5 μm can experience airborne transportation over long distances (like about 5000 km), even cross-continent from Asia to North America (Jaffe et al., 1999; Satheesh and Moorthy, 2005)."

- page 2, line 38 it should be "...CLP is expected to have important influence" instead of "...CLP will have important influence", as the statement is not proven.

Response:

Thanks for your comments. We have modified this description accordingly in the revised manuscript:

"Therefore, loess dust emitted from CLP is expected to have important influence on the radiation balance at both source areas and places far away from sources."

- page 2 and 3, literature values for scattering matrix: Please elaborate on what the scattering matrix tells us, which properties do Fij and their quotients describe? Explain either here or in section 3.1.

Response:

Thank you for the constructive comments. Scattering matrix elements describe the depolarization or transformation of incident light with several polarization states under the influences of particles. Accordingly, we have added descriptions of matrix elements in Introduction and Section 3.1 in the revised manuscript:

"Light scattering matrix $F$, a $4 \times 4$ matrix containing 16 elements $F_{ij}$ ($i, j$=1-4), is a fundamental optical property to characterize airborne dust particles, and describes the depolarization or transformation of incident light with several polarization states under the influences of particles (Quinby-Hurt et al., 2000; Volten et al., 2001)."

"Matrix elements describe the depolarization or transformation of incident light with several polarization state under the influence of particles (Quinby-Hurt et al., 2000). $F_{11}$ describes transformation of incident light intensity; $F_{12}$ describes depolarization of $0°$ and $90°$ linearly polarized light relative to scattering plane; $F_{22}$ describes transformation of $\pm 90°$ polarized incident light to $\pm 90°$ polarized scattered light and it equals to $F_{11}$ for spherical particles; $F_{33}$ and $F_{44}$ describe transformation of $\pm 45°$ linearly (or circularly) polarized incident light to $\pm 45°$ linearly (or circularly) polarized scattered light and these two elements are equal for spherical particles; $F_{34}$ describes transformation of circularly polarized incident light to $\pm 45°$ linearly polarized scattered light. Almost all these matrix elements are sensitive to physical properties of particles, including size distribution, particle shape, micro structure and refractive index."

- page 4, line 120: SEM "images", instead of "photographs", as this is an imaging technique detecting electrons, not photons.

Response:

Thank you for pointing this out. We have modified this description in the revised manuscript:

"Scanning electron microscope (SEM) images for "pristine loess" (left panel) and "milled loess" (right panel) are displayed in Figure 2."

- Table 2 and paragraph 1 on page 5: Are the differences in the sample composition significant? What are the errors on this analysis?

Response:
 Thanks for your valuable comments. We added repeat measurements of chemical compositions of each loess sample. Then, the weight percentage of each composition was averaged from three measurements, and Table 2 has been updated using averaged values and measurement errors of components. Actually, the composition differences between these two loess samples are very small. We have modified related descriptions in the revised manuscript:

"As can be seen in Table 2, the largest change of content occurs for $SiO_2$, but this change is less than 2.5 % and even smaller than the errors between repeat measurements for "pristine loess" sample, and the change of $ZrO_2$ is only about 0.03 %. It can be concluded that the composition differences between these two samples are very small, and milling process has little effect on chemical compositions for loess samples."

"**Table 2.** Chemical components of "pristine loess" and "milled loess" measured by XRF-1800."

| Components | Pristine loess (wt %) | Pristine loess error (wt %) | Milled loess (wt %) | Milled loess error (wt %) |
|---|---|---|---|---|
| $SiO_2$ | 63.8278 | 3.0237 | 66.2128 | 2.0900 |
| $Al_2O_3$ | 12.3091 | 0.3772 | 11.6487 | 0.2018 |
| $CaO$ | 9.2943 | 0.9455 | 7.8286 | 0.6450 |
| $Fe_2O_3$ | 5.5260 | 0.8817 | 5.6390 | 0.7411 |
| $K_2O$ | 3.3971 | 0.3004 | 3.3574 | 0.2358 |
| $MgO$ | 2.7536 | 0.4522 | 2.4843 | 0.2665 |
| $Na_2O$ | 1.2802 | 0.0243 | 1.3470 | 0.0214 |
| $TiO_2$ | 0.8017 | 0.0595 | 0.7939 | 0.0579 |
| $P_2O_5$ | 0.3340 | 0.0452 | 0.2549 | 0.0018 |
| $SO_3$ | 0.2370 | 0.1056 | 0.1687 | 0.0721 |
| $MnO$ | 0.1240 | 0.0294 | 0.1196 | 0.0120 |
| $ZrO_2$ | 0.0583 | 0.0104 | 0.0846 | 0.0122 |
| $SrO$ | 0.0348 | 0.0064 | 0.0299 | 0.0059 |
| $Rb_2O$ | 0.0177 | 0.0041 | 0.0174 | 0.0040 |
| $Co_2O_3$ | NT[*] | - | 0.0159 | 0.0049 |
| $Y_2O_3$ | NT[*] | - | 0.0061 | 0.0025 |

- page 6, section 3.2: Add some more detail of how the analysis was done. How many measurement iterations were performed, how are the final results derived from these?

Response:

Thank you very much for the comments. As mentioned in the first paragraph of Section 4.1, three independent measurements were conducted for each loess sample, and averaged results and their errors are obtained and shown in figures. In addition, we have added more details about measurements and data processing in the revised manuscript:

"All the matrix elements of dust samples can be determined as functions of scattering angles with the help of various combinations of orientation angles of above optical elements as shown in Table 3, which is just the same as Muñoz et al. (2010)."

"**Table 3.** Combinations of orientation angles of optical axis of all the optical elements."

| Combination | $\gamma_P$ | $\gamma_{EOM}$ | $\gamma_Q$ | $\gamma_A$ | $DC(\theta)$ | $S(\theta)$ | $C(\theta)$ |
|---|---|---|---|---|---|---|---|
| 1 | 45° | 0° | - | - | $F_{11}(\theta)$ | $-F_{14}(\theta)$ | $F_{13}(\theta)$ |
| 2 | 45° | 0° | - | 0° | $F_{11}(\theta)+F_{21}(\theta)$ | $-F_{14}(\theta)-F_{24}(\theta)$ | $F_{13}(\theta)+F_{23}(\theta)$ |
| 3 | 45° | 0° | - | 45° | $F_{11}(\theta)+F_{31}(\theta)$ | $-F_{14}(\theta)-F_{34}(\theta)$ | $F_{13}(\theta)+F_{33}(\theta)$ |
| 4 | 45° | 0° | 0° | 45° | $F_{11}(\theta)+F_{41}(\theta)$ | $-F_{14}(\theta)-F_{44}(\theta)$ | $F_{13}(\theta)+F_{43}(\theta)$ |
| 5 | 90° | -45° | - | - | $F_{11}(\theta)$ | $F_{14}(\theta)$ | $-F_{12}(\theta)$ |
| 6 | 90° | -45° | - | 0 | $F_{11}(\theta)+F_{21}(\theta)$ | $F_{14}(\theta)+F_{24}(\theta)$ | $-F_{12}(\theta)-F_{22}(\theta)$ |
| 7 | 90° | -45° | - | 45° | $F_{11}(\theta)+F_{31}(\theta)$ | $F_{14}(\theta)+F_{34}(\theta)$ | $-F_{12}(\theta)-F_{32}(\theta)$ |
| 8 | 90° | -45° | 0° | 45° | $F_{11}(\theta)+F_{41}(\theta)$ | $F_{14}(\theta)+F_{44}(\theta)$ | $-F_{12}(\theta)-F_{42}(\theta)$ |

"Multiple groups of values of measurable quantities, that is the DC component $DC(\theta)$, first harmonics $S(\theta)$ and second harmonics $C(\theta)$ of voltage signal, are recorded at every scattering angle for each combination of optical elements. The first step of data processing is to average these recorded values and get their errors. The optical platform is surrounded by black curtains to avoid the effect of environmental stray light, and background signals need to be measured and subtracted. Fluctuations of dust aerosols can be eliminated by normalizing measurements of the "detector" using $DC(30°)$ measured by the "monitor". Scattering matrix elements can be extracted from preprocessed $DC(\theta)$, $S(\theta)$ and $C(\theta)$ according to Table 3. Subsequently, $F_{11}(\theta)$ is normalized to 1 at 10° scattering angle, and the remaining matrix elements $F_{ij}(\theta)$ are normalized to $F_{11}(\theta)$ at the same angle. At last, whether measurement results of scattering matrix satisfy Cloude coherency matrix test should be examined (Hovenier and Van Der Mee, 1996). Three iterations of measurements are performed for each particle sample, the final results are average of three groups of experiments, and the errors are also calculated which contain errors during every measurement and errors for repeat measurements. Furthermore, the improved apparatus is validated using water droplets. Measured all six non-zero scattering matrix elements for water droplets can be well fitted using Mie calculation results, indicating that the measurement accuracy

of apparatus are satisfactory. For more details about the measurement principle and validation method of the apparatus, it can be referred to Liu et al. (2018)."

- page 7, section 4.1: Similar as in the introduction, it should be discussed what the physical meaning of the results are. This is partly attempted in line 199, but should be done more thoroughly.

Response:
   Thanks for your comments. To our best knowledge, there are very limited direct implications of scattering matrix elements on particle properties, because optical simulation results showed that these matrix elements are sensitive to almost all physical properties of irregular particles, like micro structure, size distribution and refractive index (Liu et al., 2015; Muinonen et al., 2007; Zubko et al., 2007). Only $F_{22}(\theta)$ equals to 1 as well as $F_{33}(\theta)$ equals to $F_{44}(\theta)$ directly imply that particles are spherical.

- page 8, lines 216-217 Please add the units of the parameters.

Response:
   Thank you very much for the comments. According to Equation (1) in the manuscript, the unit of effective radius $r_{eff}$ is μm. Refractive index is expressed as $m=n+ki$, $i$ is imaginary unit, $n$ and $k$ are real and imaginary part of refractive index respectively, and both of the two parameters are dimensionless. We have made necessary modifications in the revised manuscript:
   "As shown in Table 1, effective radii for "pristine loess" and "milled loess" are 49.40 μm and 2.35 μm, respectively. The real part of refractive index for "pristine loess" is 1.65 and that for "milled loess" is 1.70."

- page 8, lines 235-240 The description is rather vague, please make it clear you're your actual finding is.

Response:
   Thank you for your valuable comments. We have modified and re-organized the related descriptions in the revised manuscript:
   "In summary, different factors have different or similar effects on a certain matrix elements. The discrepancies in scattering matrices for "milled loess" and "pristine loess" can be mainly interpreted from the perspective of difference of effective radii, while differences in other factors such as refractive index and micro structure have relatively

small contributions, and Gaussian spheres may be promising models for simulating scattering matrix for loess dust."

\- page 10, line 302: As in page 2, line 38: Rather write "is expected to affect" or similar instead of "will affect".

Response:

Thank you for the suggestions. We have modified the description in the revised manuscript:

"Loess dust aerosols originated from CLP are expected to affect the radiation balance potentially at both source areas and downwind places far away from sources, because dust particles with different sizes can be transported over different distances."

\- page 11, paragraph 2: Please make it more clear what the scattering matrices tells us. This section is now more a summary than a conclusion.

Response:

Thanks a lot for your valuable comments. We have re-organized the related descriptions in the revised manuscript to make conclusions of this study more clear:

"Even through experimentally determined angular behaviors of scattering matrix elements for "pristine loess" and "milled loess" are similar, there are still obvious discrepancies in matrix elements. More specifically, for small "milled loess", relative phase function $F_{11}(\theta)/ F_{11}(10°)$ as well as ratios $-F_{12}(\theta)/F_{11}(\theta)$ and $F_{22}(\theta)/F_{11}(\theta)$ are smaller than that for coarse "pristine loess", while ratios $F_{33}(\theta)/F_{11}(\theta)$, $F_{34}(\theta)/F_{11}(\theta)$ and $F_{44}(\theta)/F_{11}(\theta)$ are larger than that for coarse "pristine loess". These discrepancies are unique and different from that for other kinds of dust with distinct size distributions published in literatures. Qualitative analyses of optical simulations of various morphological model showed that the large difference in size distributions (effective radii differ by more than 20 times) caused by milling process plays a major role in leading to discrepancies in scattering matrices for these two samples, while differences in factors such as refractive index and micro structure have relatively small and recessive contributions. And Gaussian sphere models may have good application prospect in optical modeling of loess dust, while more detailed quantitative verification using measured physical properties are still needed."

"Synthetic scattering matrices for both "pristine loess" and "milled loess" were defined over $0°$-$180°$ scattering angle, and the previously presented average scattering matrix for loess was updated with new coarse "pristine loess" sample included. The phase function $F_{11}(\theta)$ in updated average matrix has larger forward scattering peaks and smaller values at side and backward scattering angles than that in previous average matrix. Compared to previous average matrix, updated average matrix has larger -

$F_{12}(\theta)/F_{11}(\theta)$ at side scattering angles, has smaller $F_{33}(\theta)/F_{11}(\theta)$ and $F_{44}(\theta)/F_{11}(\theta)$ at backscattering angles. $F_{22}(\theta)/F_{11}(\theta)$ experiences the largest change before and after update, whose values are enlarged at almost all scattering angles."

"In this study, scattering matrices for Chinese loess samples with large difference in their size distributions are investigated. Based on all the measurements, suitable shape distributions of spheroids can be obtained respectively, which are useful for the retrievals of airborne loess dust properties at both source and downwind areas in China or even East Asia. On the other hand, the updated average scattering matrix for loess are meaningful for the validation of exiting models and the development of more advanced morphological models suitable for loess dust, which are also useful to finally improve the retrieval accuracies of dust aerosol properties."

- page 11, line 323: Data availability: You uploaded the data, which is great, this should be linked here.

Response:

   Thank you for this suggestion. During the revision of the manuscript, we re-measured scattering matrices for the two loess samples over angles from 5 ° to 175 ° using an improved apparatus (the previous apparatus can only cover 5-160 °). Newly measured scattering matrices were in good agreement with measurement results using the previous apparatus in the range of 5-160 °. The extension of scattering angles made the polynomial extrapolation of matrix elements $F_{11}(\theta)/F_{11}(10\,°)$ and $F_{22}(\theta)/F_{11}(\theta)$ at backscattering angles more rigorous when constructing synthetic matrix, and calculated backscattering depolarization ratios were also more reliable. Accordingly, we re-uploaded measured results to a new dataset. We have attached the link of new dataset in the revised manuscript:

   "All the data involved in this study are available online at: https://github.com/liujia93/Scattering-matrix-for-loess-dust."

- Table 1: Add units.

Response:

   Thank you for the comments. As mentioned above, the unit of effective radius $r_{eff}$ is μm, real part $n$ and imaginary part $k$ of refractive index are dimensionless. According to Equation (2), the effective standard deviation $\sigma_{eff}$ is dimensionless. In addition, since the effective size parameter $x_{eff}$ is defined as $x_{eff}=2\pi r_{eff}/\lambda$, this parameter is also dimensionless and has no units.

- Abstract and Conclusions: Please add: What do the results of that study actually tell us about light scattering by Chinese loess in one sentence?

Response:
   Thanks a lot for your valuable comments. As can be seen from the Response to the last General Comments, we summarized three major results and conclusions of our study. It is hard to describe the conclusions using just one sentences, but we have modified related descriptions in Abstract and Conclusions in the revised manuscript.
   "Experimental results showed that there are obvious discrepancies in angular behaviours of matrix elements for "pristine loess" and "milled loess", and these discrepancies are different from that for other kinds of dust with distinct size distributions. Given that the effective radii of these two loess samples differ by more than 20 times, it is reasonable to conclude that the difference in size distributions plays a major role in leading to different matrices, while differences in refractive index and micro structure have relatively small contributions. Qualitative analyses of numerical simulation results of irregular particles also validate this conclusion. Gaussian spheres may be promising morphological models for simulating scattering matrix of loess but need further quantitative verification."
   "This study presents measurement results of Chinese loess dust and updated average scattering matrix for loess, which are useful for validating existing models and developing more advanced models for optical simulations of loess dust and finally help to improve retrieval accuracy of dust aerosol properties over both source and downwind areas."
   "Even through experimentally determined angular behaviors of scattering matrix elements for "pristine loess" and "milled loess" are similar, there are still obvious discrepancies in matrix elements. More specifically, for small "milled loess", relative phase function $F_{11}(\theta)/ F_{11}(10°)$ as well as ratios $-F_{12}(\theta)/F_{11}(\theta)$ and $F_{22}(\theta)/F_{11}(\theta)$ are smaller than that for coarse "pristine loess", while ratios $F_{33}(\theta)/F_{11}(\theta)$, $F_{34}(\theta)/F_{11}(\theta)$ and $F_{44}(\theta)/F_{11}(\theta)$ are larger than that for coarse "pristine loess". These discrepancies are unique and different from that for other kinds of dust with distinct size distributions published in literatures. Qualitative analyses of optical simulations of various morphological model showed that the large difference in size distributions (effective radii differ by more than 20 times) caused by milling process plays a major role in leading to discrepancies in scattering matrices for these two samples, while differences in factors such as refractive index and micro structure have relatively small and recessive contributions. And Gaussian sphere models may have good application prospect in optical modeling of loess dust, while more detailed quantitative verification using measured physical properties are still needed."

References
Liu, J., Yang, P., and Muinonen, K.: Dust-aerosol optical modeling with Gaussian spheres: Combined invariant-imbedding T-matrix and geometric-optics approach,

Journal of Quantitative Spectroscopy and Radiative Transfer, 161, 136-144, http://doi.org/10.1016/j.jqsrt.2015.04.003, 2015.

Liu, J., Zhang, Y., and Zhang, Q.: Laboratory measurements of light scattering matrices for resuspended small loess dust particles at 532 nm wavelength, Journal of Quantitative Spectroscopy and Radiative Transfer, 229, 71-79, https://doi.org/10.1016/j.jqsrt.2019.03.010, 2019.

Muinonen, K., Zubko, E., Tyynelä, J., Shkuratov, Y. G., and Videen, G.: Light scattering by Gaussian random particles with discrete-dipole approximation, Journal of Quantitative Spectroscopy and Radiative Transfer, 106(1-3), 360-377, doi:10.1016/j.jqsrt.2007.01.049, 2007.

Zubko, E., Muinonen, K., Shkuratov, Y., Videen, G., and Nousiainen, T.: Scattering of light by roughened Gaussian random particles, Journal of Quantitative Spectroscopy and Radiative Transfer, 106(1-3), 604-615, doi:10.1016/j.jqsrt.2007.01.050, 2007.

---

## Author Comment (AC4) · 14 Jun 2020

**Responses to the comments of reviewer 4**

The authors really appreciate the valuable comments and constructive suggestions from the reviewer. The suggestions and comments of reviewer are listed in black font, and responses are highlighted in blue. The changes made in the revised manuscript are marked in red font.

**Comments from reviewer 4:**

The study presents light scattering measurements of Chinese loess dust. The authors have measured the scattering matrix elements of a single loess sample from the Chinese loess plateau once untreated (pristine loess) and once milled (milled loess) and performed some complementary measurements too. I find the topic very interesting and useful. However, I have some doubts about the paper being published in its current form.

Response:
   Thank you very much for reviewing our manuscript and all these valuable comments and suggestions. We have tried our best to respond your comments point by point and modified related descriptions in the revised manuscript. And we hope that you will reconsider our manuscript.

First of all I have to question if the choice of the journal is adequate for the performed study. I might be wrong about this and if this is the case, then please just ignore this comment. However, this journal is called Atmospheric Measurement Techniques, and on its homepage it is stated that: "The main subject areas comprise the development, intercomparison, and validation of measurement instruments and techniques of data processing and information retrieval for gases, aerosols, and clouds." This paper presents none of them. It shows some laboratory measurements with atmospheric relevance. It does not show a new measurement technique nor a newly developed instrument neither any instrument intercomparison. The only technical part of the paper is the one page section of 3.2 where the measurement apparatus is shortly introduced.

Response:
   Thanks a lot for your comments.
   We think our work can be classified into subject areas "techniques of information retrieval for aerosols". Accurate retrievals of optical and physical properties of dust aerosols depend largely on the choice of suitable particle models of dust. So the model development for dust has always been worthy of attention, and we think non-spherical particle models for dust particles are still needed to be further verified or developed targeting for specific kinds of dust with different physical properties.
   Chinese loess dust contributes a lot to Asian dust and is expected to affect radiative balance over both source and downwind regions. However, there is still no specific particle models for Chinese loess dust. Therefore, in our study, scattering matrices and essential physical properties of Chinese loess dust samples with different size

distributions, which represent dust aerosols over source and downwind regions respectively to some extent, are investigated. All these measurement results are necessary constraints for the development of advanced particle models or the retrievals of best fitted shape distributions of widely used spheroid models (Dubovik et al., 2006). And we believe that these models will help to improve the retrieval accuracy of physical properties of Chinese loess dust aerosols over both source and downwind regions. Furthermore, the updated average scattering matrix for loess is also instructive to the model development of loess dust and useful for improving the retrieval accuracy of dust aerosol properties over other loess regions in the world.

My other main concern is: if the manuscript contains strong enough scientific material to be published in AMT. The scattering matrix element measurements of the two differently treated loess sample come from 6 single measurements, and the manuscript is based completely on this. It would considerably improve the manuscript if more measurements were included. To give you some ideas: include measurements and a comparison of different kind of loess samples collected either on the Chinese Loess Plateau at other places or get loess samples from outside of China. Another idea could be to include some other types of mineral dust and make a comparison. I know well, that it is not always possible to perform more measurements additionally. The manuscript could be improved with much thorougher discussion about comparing existing literature data with your dataset as well, or perform some numerical simulations based on the measured size distribution and shape (e.g. Mie theory and a theory for non-spherical particles) and discuss the results.

Response:
   Thank you for the comments and suggestions.
   In this study, we experimentally investigated scattering matrices as well as other basic physical properties of Chinese loess dust with two distinct size distributions, from a meaningful perspective of long range transport of dust particles. Furthermore, we explored reasons for the discrepancies in scattering matrices based on qualitative analyses of optical simulations in literatures, and updated the previous average scattering matrix for loess dust. All the discussions are focused on the loess dust.
   Until now, experimental studies of scattering matrices are still very rare. Before our series of studies, only one Hungary loess sample was characterized at 441.6 and 632.8 nm wavelengths by Volten et al. (2001). In our previous study, fine loess particles sampled from two typical regions of Chinese Loess Plateau were investigated at 532 nm wavelength and compared with measurement results of Hungary loess (Liu et al., 2019). Comparisons showed that measured scattering matrices for different samples have good consistencies, thus an average scattering matrix for loess dust was built. The average scattering matrix for loess published in our previous study (Liu et al., 2019) is called as "previous average scattering matrix" in the current study, and we updated it using new coarse "pristine loess" sample. Therefore, in other words, the differences between average matrix before and after update are also the differences between "pristine loess" and the other three samples, and differences among these three samples can be referred to Liu et al. (2019). As can be seen in Figure 6 in the manuscript, compared to other three samples, phase function for "pristine loess" has larger forward scattering peaks and smaller values at side and back scattering directions. "Pristine loess" has larger $-F_{12}/F_{11}(\theta)$ values at near side scattering angles, has larger $F_{22}/F_{11}(\theta)$ values

at almost all scattering angles, and has smaller values of both $F_{33}/F_{11}(\theta)$ and $F_{44}/F_{11}(\theta)$ at backscattering angles, when compared with the other three samples.

As for discrepancies in scattering matrices among different kinds of mineral dust, Volten et al. (2001) had already made such comparisons, these discrepancies are obvious, but it is still very hard to discriminate dust types using angular distributions of matrix elements. This is because physical properties such as size distribution, refractive index and micro structure are all different to some degree, and direct and rough comparison may be not so meaningful. In our another previous work (Liu et al., 2018), we compared scattering matrices for anthropogenic cement dust with that for natural mineral dust, it is also difficult to discriminate one certain dust type from others based on scattering matrices. We did not made such analysis in our manuscript, because the average scattering matrix for loess may also cover measurement results of other dust types, and this will confuse readers. And there is an essential underlying premise for the application of average scattering matrix for loess dust, that is the tracing of airborne dust using models like HYSPLIT Model ensures its source is loess regions.

Many studies had shown that Mie calculations of spheres cannot reproduce measured scattering matrices for mineral dust at all (Meng et al., 2010; Merikallio et al., 2015; Mishchenko et al., 2003). Therefore, we did not make direct comparisons between Mie calculations and measured scattering matrices, because no more information can be extracted except that non-spherical shape of loess. In contrast, we are more prefer to conduct optical modeling with non-spherical models. However, optical modeling of irregular dust particles with large sizes is still a very challenging subject, and only few researcher focus on it. We had tried to contact these experts for cooperation, but didn't get any response. We want to attract interest of modeling experts by presenting some meaningful experimental results, and we hope to establish cooperation with these experts in future, since only combinations of experiments and optical simulations can make our work more complete and useful. Even so, in the subsection "4.1 Experimentally Determined Scattering Matrices", we tried to find the main factor that resulting in these distinctions in measured scattering matrices for two loess samples based on qualitatively analyses of numerical simulations, simulation results of non-spherical Gaussian particles and agglomerated debris particles were selected for analyses. And we found that Gaussian spheres with effective radii same as measured size distribution of loess samples can qualitatively explain these measured distinctions in scattering matrices.

We have modified related descriptions about comparisons between previous and updated average scattering matrix in revised manuscript:

"At last, the previously published average scattering matrix for loess, which consists of results for Hungary loess, milled Yangling loess and milled Luochuan loess (the latter two were sampled from CLP), was updated using new sample "pristine loess" from Luochuan, by averaging synthetic matrices for different loess samples. In other words, the differences between average matrix before and after update are also the differences between "pristine loess" and the other three samples, and differences among these three samples can be referred to Liu et al. (2019). As shown in Figure 6, compared to other three samples, phase function for "pristine loess" has larger forward scattering peaks and smaller values at side and back scattering directions. "Pristine loess" has larger -$F_{12}(\theta)/F_{11}(\theta)$ values at near side scattering angles, has larger $F_{22}(\theta)/F_{11}(\theta)$ values at almost all scattering angles, and has smaller values of both $F_{33}(\theta)/F_{11}(\theta)$ and $F_{44}(\theta)/F_{11}(\theta)$ at backscattering directions, when compared with the other three samples."

You could also improve the paper by stating clearly what your main message is for the reader. You just present the scattering matrix elements but do not draw any further conclusions. How is Chinese loess scattering treated in radiative transfer models? Will there be a big difference if these models are updated with your results? How representative is your single loess sample?

Response:

Thanks for your valuable comments.

The main messages for readers in this study can be concluded as three aspects: (1) there are obvious discrepancies in measured scattering matrices for Chinese loess dust with different size distributions, and these discrepancies are different from that for other kinds of mineral dust with various size distributions. (2) Qualitative analyses of numerical simulation results in literatures showed that the large difference in size distributions (effective radii differ by more than 20 times) plays a major role in leading to these discrepancies in scattering matrices. And Gaussian spheres may be promising models for simulating scattering matrix for Chinese loess dust, but more detailed quantitative verifications using measured size distributions and refractive indices are still needed. (3) The previously published average scattering matrix for loess dust was updated using measurements of new coarse loess sample, which is meaningful for validating existing models and developing more advanced models suitable for optical simulations of loess dust, and finally helps to retrieve dust aerosol properties with higher accuracy over both source and downwind areas.

To our best knowledge, there is no study focus on the selection of optical model for Chinese loess in radiative transfer models. As for optical models for mineral dust, Dubovik et al. (2006) used simulated scattering matrices of spheroid models with different aspect ratios to reproduce measured results for different kinds of mineral dust published by Volten et al. (2001), and a best-fitted shape distribution of spheroids was recommend. Subsequent studies on the retrievals of dust aerosol properties from space-based (Dubovik et al., 2011), airborne (Espinosa et al., 2019) and ground-based (Titos et al., 2019) remote sensing observations were conducted based on this shape distribution. Tian et al. (2019) retrieved the total aspect ratio distributions of spheroids for all kinds of aerosols over Chinese Loess Plateau from depolarization ratios observed by lidars, however, aspect ratio distributions for loess dust still cannot be separated. Furthermore, optical simulations and radiation transfer calculations conducted by Li et al. (2019) showed that shape distributions of spheroids have obvious effects on scattering matrices and further affect radiance distribution and polarization properties of sky light. Therefore, we think the best fitted shape distributions of spheroids for loess dust with distinct sizes are still highly in demand, and the accuracy of retrieved dust aerosol properties will be further improved with the help of these best fitted models.

As mentioned in manuscript, original loess sample was collected from Luochuan Loess National Geological Park, the only national park for loess landform in China. So this sample represents Chinese loess to some extent, but it cannot represent all loess distributed in China. In our previous work (Liu et al., 2019), we investigated fine loess particles sampled from Luochuan and Yangling, the latter located at the southern edge of Chinese Loess Plateau. Even through measured scattering matrices have good consistencies, there are still obvious discrepancies in the angular distributions of matrix elements, and these discrepancies even larger than the differences between loess with different size distributions in this study. This means local variations of loess also have significant effect on scattering matrix.

The measurement results of all these loess samples were included in the average scattering matrix for loess, however, these samples may still cannot represent all loess in China, even all loess in Chinese Loess Plateau. Therefore, we will further update the average scattering matrix for loess dust in future using measurements of more samples collected from different regions of China and more samples with different sizes.

We have added necessary descriptions in revised manuscript:

"Fine loess dust sampled from Luochuan and Yangling, two regions of Chinese Loess Plateau, were investigated by Liu et al. (2019). Local variations of loess dust also have obvious effects on the measured scattering matrices. It should be noted that all these samples investigated may still cannot completely represent the loess in Chinese Loess Plateau and China, so one of the efforts in the future is to investigate more loess samples collected from more regions and with more size distributions, accordingly, the average scattering matrix for loess will be updated constantly."

You probably cannot implement all of my main suggestions to improve the manuscript, and it is not necessary either. I just wanted to show you some possible options how it could be done. The data and the work you do is valuable but I only can recommend the manuscript's publication if it is significantly improved.

Response:

Thank you very much for all these meaningful comments and suggestions. We have tried our best to response the comments and revise our manuscript. We also explained the reasons for some of your suggestions cannot be implemented right now, and listed them as our future works. And we hope that you can re-consider our manuscript.

Other Comments:
1. I suggest a careful English language editing of the manuscript.

Response:

Thanks for your suggestion. We have tried our best to correct language mistakes by repeatedly reviewing the manuscript, and we also have invited native speakers to edit the manuscript.

2. Page 4, Lines 99-111: Even after a longer search I could not find details about the SALD-2300 instrument and how it exactly measures the particle size distribution and refractive index. Please add details how it exactly works. What I think it does is measuring the light scattering at many angles and trying to reproduce the measurement with a guessed number size distribution and a refractive index using theoretically calculated scattering values. Does it use the Mie theory (which is valid for spherical particles only)? Or how can it calculate the scattering for particles with unknown shape? How does it influence your derived number size distribution and refractive index? What is the uncertainty of this measurement method for non-spherical particles? Please add a discussion on this. Are you sure that the refractive index difference between 1.65+0i for "pristine loess" and 1.70+0i for "milled loess" is real?

Response:

Thank you very much for the comments.

As you point out, SALD-2300 measures the angular distribution of scattered light intensity, then many combinations of values of number size distribution and refractive index are employed for Mie calculations to reproduce the measured light intensity distribution, the best fitted size distribution and refractive index of sample are obtained at last. SALD-2300 has 84 light detectors in all, including 78 forward detector elements, one side detector and five back detectors. Liu et al. (2003) revealed that Mie theory can be used to reproduce forward scattering intensities of nonspherical particles with moderate aspect ratios at scattering angles smaller than 20 °. Since over 70% of the detectors of SALD-2300 are set at angles smaller than 20 °, so we think the retrieved size distributions of nonspherical loess dust are of high accuracy. It is hard to further evaluate uncertainty of measured size distribution, because there is still no complex model for loess dust suitable for its optical simulation, which is the final goal of our work.

The smallest calculation steps of real and imaginary part of refractive index for SALD-2300 is 0.05 and 0.01, and these two values are chosen to retrieve refractive index for loess samples. All three repeat measurements obtained the same refractive indices for both "pristine loess" and "milled loess". Kinoshita (2001) retrieved refractive indices for alumina dust with different sizes using the same method as SALD-2300, there were also small difference 0.05 in the retrieved real part, and he explained this phenomenon as the effect of nonspherical property of dust. So we think there is indeed difference in the retrieved refractive indices for the two loess dust samples with distinct size distributions. And this is also due to the nonspherical nature of loess particles. From the perspective of numerical simulation, the effect of refractive index on angular distribution of scattered light intensity of nonspherical complex particles are still unclear enough (Muinonen et al., 2007; Zubko et al., 2013). Furthermore, to our best knowledge, there is still no certain conclusion of refractive index of loess, so it is hard to evaluate the uncertainty of our retrieved results.

We have added more detailed descriptions about SALD-2300 in the revised manuscript:

"SALD-2300 has 84 scattering light detectors in all, including 78 forward detector elements, one side detector and five back detectors. The best fitted number size distribution and refractive index $m$ can be obtained by reproducing measured angular distribution of light intensity based on Mie calculations. Liu et al. (2003) revealed that Mie theory can be used to reproduce forward scattering intensities of nonspherical particles with moderate aspect ratios at scattering angles smaller than 20 °. Since over 70% of the detectors of SALD-2300 are set at angles smaller than 20 °, the retrieved size distributions of nonspherical loess dust based on Mie theory are of relatively high accuracy. During size distribution measurements of loess samples, the retrieval ranges of real part $Re(m)$ and imaginary part $Im(m)$ of refractive index were preset as 1.45-1.75 and 0-0.05, respectively (Volten et al., 2001). The smallest calculation steps of $Re(m)$ and $Im(m)$ are 0.05 and 0.01, respectively."

3. Page 4, Lines 110-111: "larger particles have relatively larger real part of refractive index": if I understood correctly your method of producing the milled loess sample, it contains exactly the same material (your chemical analysis verifies it) as the pristine

loess and therefore one would expect the two samples having the same refractive index. Are you sure, again, that your result is real and are not only a measurement artifact/uncertainty? Or do you think that the milling caused some strange structural changes in the loess sample which homogenized or inhomogenized how the chemical components are distributed within a single particle and/or between the particles?

Response:

Thanks a lot for the comments.

First of all, there is a clerical error in this sentence, it should be "larger particles have relatively smaller real part of refractive index". There is no doubt that refractive index of specific material is unique. And we don't think the milling process obviously modified the distribution of chemical components within a single and between the particles.

Retrieved refractive index of particles based on measured light intensity distribution is a kind of optically equivalent refractive index, it is close to inherent refractive index of the measured material but not necessarily the same. Kinoshita (2001) retrieved refractive indices for alumina dust samples with 1 μm and 5 μm diameter. The inherent refractive index of alumina is known as 1.76, while the retrieved real parts are 1.80 and 1.75 respectively, larger particles have smaller real part of refractive index and the difference is small but cannot be ignored. Kinoshita explained this phenomenon as the effect of nonspherical nature. Our study also found larger particles have slightly smaller real part of refractive index, so we think this difference is real and can be explained by the same reason.

We have modified the mentioned clerical error and added necessary discussions in revised manuscript:

"As shown in Table 1, the optimal refractive indices are 1.65+0$i$ for "pristine loess" and 1.70+0$i$ for "milled loess", larger particles have relatively smaller real part of refractive index, which is similar to the results of Kinoshita (2001) and is caused by the nonspherical nature of loess dust. Retrieved refractive index of particles based on measured light intensity distribution is a kind of optically equivalent refractive index, which is close to the inherent refractive index of the measured particles."

4. Page 6, Section 3.2: I assume that this is not the first paper which uses this experimental apparatus. Please add a reference to the paper where a more detailed description of your instrument is available. If there is no such paper, please add a more detailed description.

Response:

Thank you for pointing this out. During the revision of the manuscript, we improved the experiment apparatus by extending the maximum angle coverage from 160 °to 175 °, the photograph of improved apparatus are shown below, and re-measured scattering matrices for the two loess samples.

[Figure]

[Figure]

Newly measured scattering matrices were in good agreement with measurement results using the previous apparatus in the range of 5-160°. As we mentioned in the original manuscript: "For more details, it can be referred to Muñoz et al. (2010) and Liu et al. (2018)." We are sorry that this description may be not clear enough and confusing. Therefore, we have modified the confused descriptions and added more details of the improved apparatus in the revised manuscript:

"The main improvement is that angle coverage at backscattering angles are extended to 175°, while the maximum coverage of previous apparatus is 160°(Liu et al., 2018)."

"The dark cassette used to encapsulate the "detector", *Q* and *A* in previous apparatus is removed, which facilitate the adjustment of orientation angles of *Q* and *A*."

"Furthermore, the improved apparatus is validated using water droplets. Measured all six non-zero scattering matrix elements for water droplets can be well fitted using Mie calculation results, indicating that the measurement accuracy of apparatus are satisfactory. For more details about the measurement principle and validation method of the apparatus, it can be referred to Liu et al. (2018)."

5. Page 6, Lines 59-62: Since your main results are the measured matrix elements, probably it would be worth explaining exactly from which polarization states which matrix elements were derived and how, and not only referencing a paper for it.

Response:

Thanks a lot for your suggestions. We have add a new table and more details about the relationship of combinations of optical elements and matrix elements in the revised manuscript:

"All the matrix elements of dust samples can be determined as functions of scattering angles with the help of various combinations of orientation angles of above optical elements as shown in Table 3, which is just the same as Muñoz et al. (2010)."

"**Table 3.** Combinations of orientation angles of optical axis of all the optical elements."

| Combination | $\gamma_P$ | $\gamma_{EOM}$ | $\gamma_Q$ | $\gamma_A$ | $DC(\theta)$ | $S(\theta)$ | $C(\theta)$ |
|---|---|---|---|---|---|---|---|
| 1 | 45° | 0° | - | - | $F_{11}(\theta)$ | $-F_{14}(\theta)$ | $F_{13}(\theta)$ |
| 2 | 45° | 0° | - | 0° | $F_{11}(\theta)+F_{21}(\theta)$ | $-F_{14}(\theta)-F_{24}(\theta)$ | $F_{13}(\theta)+F_{23}(\theta)$ |
| 3 | 45° | 0° | - | 45° | $F_{11}(\theta)+F_{31}(\theta)$ | $-F_{14}(\theta)-F_{34}(\theta)$ | $F_{13}(\theta)+F_{33}(\theta)$ |
| 4 | 45° | 0° | 0° | 45° | $F_{11}(\theta)+F_{41}(\theta)$ | $-F_{14}(\theta)-F_{44}(\theta)$ | $F_{13}(\theta)+F_{43}(\theta)$ |
| 5 | 90° | -45° | - | - | $F_{11}(\theta)$ | $F_{14}(\theta)$ | $-F_{12}(\theta)$ |
| 6 | 90° | -45° | - | 0 | $F_{11}(\theta)+F_{21}(\theta)$ | $F_{14}(\theta)+F_{24}(\theta)$ | $-F_{12}(\theta)-F_{22}(\theta)$ |
| 7 | 90° | -45° | - | 45° | $F_{11}(\theta)+F_{31}(\theta)$ | $F_{14}(\theta)+F_{34}(\theta)$ | $-F_{12}(\theta)-F_{32}(\theta)$ |
| 8 | 90° | -45° | 0° | 45° | $F_{11}(\theta)+F_{41}(\theta)$ | $F_{14}(\theta)+F_{44}(\theta)$ | $-F_{12}(\theta)-F_{42}(\theta)$ |

"Multiple groups of values of measurable quantities, that is the DC component $DC(\theta)$, first harmonics $S(\theta)$ and second harmonics $C(\theta)$ of voltage signal, are recorded at every scattering angle for each combination of optical elements. The first step of data processing is to average these recorded values and get their errors. The optical platform is surrounded by black curtains to avoid the effect of environmental stray light, and background signals need to be measured and subtracted. Fluctuations of dust aerosols can be eliminated by normalizing measurements of the "detector" using $DC(30°)$ measured by the "monitor". Scattering matrix elements can be extracted from preprocessed $DC(\theta)$, $S(\theta)$ and $C(\theta)$ according to Table 3. Subsequently, $F_{11}(\theta)$ is normalized to 1 at 10° scattering angle, and the remaining matrix elements $F_{ij}(\theta)$ are normalized to $F_{11}(\theta)$ at the same angle. At last, whether measurement results of scattering matrix satisfy Cloude coherency matrix test should be examined (Hovenier and Van Der Mee, 1996). Three iterations of measurements are performed for each particle sample, the final results are average of three groups of experiments, and the errors are also calculated which contain errors during every measurement and errors for repeat measurements."

6. Page 7, Line 193: "all six non-zero matrix elements are limited to narrow regions, respectively" I don't understand what you mean here. What narrow regions? Angle range? Y-value range? Or do you mean that your error bars are small? Please clarify!

Response:
   Thank you very much for pointing this out. What we want to say in this sentence is that matrix elements for both "pristine loess" and "milled loess" present similar angular behaviors. More specifically, angular distributions of all six non-zero matrix elements, in other words Y-values in each sub plot, are limited to narrow regions, respectively. We have modified these confused descriptions in the revised manuscript:
   "Matrix element ratios for "pristine loess" and "milled loess" present similar angular behaviors, more specifically, angular distributions of all six non-zero matrix element ratios are limited to narrow regions, respectively."

7. Page 7, Lines 199-201: Please comment on the angular behavior of F_22. Next to it: it looks like that the milled loess sample deviates more from unity than the pristine loess sample. Does this suggest that the milled loess has a more irregular shape than pristine loess?

Response:
  Thanks a lot for your valuable comments. As we mentioned in manuscript, $F_{22}(\theta)/F_{11}(\theta)$ equals to constant 1 when particles are homogeneous spheres, otherwise, particles are nonspherical and irregular. However, this does not mean that different $F_{22}(\theta)/F_{11}(\theta)$ measurement results can directly indicate the discrepancy of particle irregularity. Because optical simulations of nonspherical Gaussian particles conducted by Liu et al. (2015) showed that $F_{22}(\theta)/F_{11}(\theta)$ values are not only sensitive to particle irregularity but also to particle size. We have added necessary discussions in the revised manuscript:
  "Experimentally determined $F_{22}(\theta)/F_{11}(\theta)$ values of "milled loess" are larger than "pristine loess", especially at side and back scattering angles. It should be noted that discrepancies in measured $F_{22}(\theta)/F_{11}(\theta)$ cannot be directly used to indicate difference of particle irregularity, because optical calculations of Gaussian spheres showed that $F_{22}(\theta)/F_{11}(\theta)$ values are sensitive to not only particle irregularity but also to size distribution (Liu et al., 2015)."

8. Page 7, Lines 206-207: The sentence is very confusingly phrased, please rephrase it. I am not sure if I understood what you wanted to tell the reader but I don't see any significant difference between the 5 °relative phase functions.

Response:
  Thank you for pointing this out. There is a drawing error in subplot $F_{11}/F_{11}(10°)$, we are very sorry about that. In the original manuscript, the display range of $F_{11}/F_{11}(10°)$ was set as 0.005-10 in log scale, but relative phase function at 5° for "pristine loess" is about 15, this can be checked from the dataset previously published by us at https://doi.org/10.5281/zenodo.3361852. We have corrected this error by resetting the maximum display value as 20 in subplot $F_{11}/F_{11}(10°)$ in revised manuscript:
  "On the other hand, the discrepancies in matrix elements for "pristine loess" and "milled loess" are still obvious. Compared to "milled loess", there is an enlargement of relative phase function at 5° scattering angle for "pristine loess"."

9. Page 7, Lines 206-208: From F11 it looks like that milled loess has a higher forward to backward scattering ratio than pristine loess. I would expect exactly the other way around because the pristine loess sample contains much larger particles and larger particles usually have a much higher forward scattering compared to the backward scattering value. Please comment on it.

Response:
  Thanks a lot for you valuable comments. Similar to the response to Comment 8, we are sorry there is an error in subplot $F_{11}/F_{11}(10°)$. And we have corrected it in the revised manuscript. We agree that larger particles have much higher ratios of forward

scattering to backward scattering. In our study, the forward (5 °) to backward scattering (175 °) ratio of "pristine loess" is about 3.60 times larger than that of "milled loess".

10. Page 8, Lines 119-223: Is there no way to produce samples containing smaller particles than the original without changing their form? Just by sieving the sample (the size distribution of the pristine loess seems to me broad enough)? Would that not work? If it would, then measuring such samples could save you from speculating about, if the measured differences are due to the different size or shape. It would be also very nice to have more samples with different sizes and not only two. You show that the particle size differs much more than the shape between the two samples, and your speculations might be true as well. However, how can you be sure that every component of the scattering matrix is comparably sensitive to the changes in size and shape? Let's assume, that one matrix component is 1000 times more sensitive to the changes in the particle shape than to the changes in the size? Please provide some proof that such a case is not to be expected, and then your argumentation becomes valid.

Response:
  Thank you very much for your meaningful comments.
  During our sample preparation stage, we did try to use 20 μm and 10 μm sieves to obtain loess samples with different size distributions, but we ended in failure. Only very few particles were obtained using 20 μm sieve, which is far from meeting the requirement of light scattering matrix measurements, and there were almost no particles can be obtained using 10 μm sieve. We also did not find other available methods that can be used to prepare enough particles for experiments based on the limited original loess samples, so we use ball milling method. We think ball milling can modified particle shapes to some extent, but we also have a question that whether the particles with different sizes in original loess sample can be described by the same morphology, since, to our best knowledge, there is still no effective method to adequately describe real morphologies of irregular dust particles using several parameters.
  For loess particles with effective radii smaller than "pristine loess" but larger than "milled loess", it can be summarized from optical simulations of Gaussian spheres that both size and irregularity have roughly similar effects on matrix element ratios $F_{33}/F_{11}$, $F_{34}/F_{11}$ and $F_{44}/F_{11}$, so it is almost impossible to tell which of the two factors plays a major role (Liu et al., 2015). In such cases, qualitative analysis is far from enough, only quantitative analysis in cooperation with optical model experts can separate the effects of size and irregularity. So we did not investigate samples with effective radii between "pristine loess" and "milled loess" in this study, and we want to investigate such samples in combination with quantitative optical simulations by cooperating with optical modeler, only in this way can our research more meaningful.
  As far as we know, it is hard to evaluate the sensitivity of each scattering matrix element to the changes of size and irregularity, because the effects of these two factors are usually both complex and even coupled. Another reason is that it is very hard to use morphological models to adequately describe real dust particles, so the determination of variation ranges of model morphological parameters is hard, and then it is hard to assess the effects of size and irregularity using the same relative change standard. Liu et al. (2015) calculated scattering matrices for Gaussian spheres whose size parameter ranges from 1 to 1000 and standard deviation of radial distance (irregularity) ranges from 0 to 0.2. The effective size parameters of "pristine loess" and "milled loess" are

about 580 and 30 respectively. As this parameter increases from 30 to 580, there are significant variations in matrix elements, the variations of $F_{11}$, $-F_{12}/F_{11}$, $F_{33}/F_{11}$ and $F_{44}/F_{11}$ are more obvious than the effect of irregularity increasing from 0.05 to 0.2, the sensitivity differences of $F_{22}/F_{11}$ and $F_{34}/F_{11}$ on size and irregularity are definitely less than 1000 times. It should be noted that the comparisons are not rigorous enough, because the irregularity range 0.05-0.2 may not exactly applicable to our loess samples. On the other hand, commercial laser particle size analyzers such as SALD-2300 employ Mie theory to retrieve size distribution of irregular dust particles based on light intensity distribution (matrix element $F_{11}$), this is because $F_{11}$ is more sensitive to size than particle irregularity, especially in forward scattering angles, otherwise these instruments cannot be used to measure particle size at all.

11. Page 8, Lines 224-240: It would considerably strengthen the manuscript if numerical calculations based on your measured size distribution and particle shape were added and not only the existing literature was analyzed. If that is not possible, you should show how the size and shape of your samples compare to the size and shape of the particle in the referenced papers. Irregular dust does not necessary mean comparable size distribution and/or particle shape.

Response:
   Thanks a lot for the valuable comments.
   The morphology of irregular dust particle is difficult to be adequately described by several parameters, even the most advanced models cannot always correspond to real particles directly. Until now, researches on optical simulations of irregular particles are still very rare. Simulations of agglomerated debris model and rough Gaussian model only cover very small size range of dust particles and calculations of larger particles are very time-consuming (Zubko et al., 2007, 2013). Therefore, the influence of morphological parameters such as size and irregularity can only be roughly summarized and extended to large particles. However, optical simulations of Gaussian sphere model cover most of the particle size distributions of our loess samples, so they are used for direct qualitative analyses in our study (Liu et al., 2015).
   Furthermore, quantitative analyses can only be performed using measured particle size distributions, morphological parameters are very hard to be taken into consideration. The most advanced optical modeling method for dust particles can only employ a few irregular shape models with specific morphological parameters (which cannot correspond to real dust particles directly), and the measured particle size distributions are employed. Then, calculation results were used to reproduce the measured scattering matrix, the best fitted number fractions of irregular shape models mentioned above (shape distributions) can be retrieved finally, and these shape distributions represent particle irregularity to some extent. Optical modeling of irregular dust is still an urgent and challenging problem. In future, we hope to combine experimental measurements and optical simulations of models much closer to real morphology of dust by cooperating with optical modeling experts to make our investigations more meaningful.

12. Page 9, Section 4.2: During calculating the synthetic scattering matrices you follow the works of Dabrowska et al., 2015 and Escobar-Cerezo et al., 2018. They used the very same measurement technique, had only different kind of samples (Lunar and Martian dust). You clearly follow their work, by extrapolating the measurements to the angles you could not measure as well. The extrapolation in the forward direction is based on the Mie theory and is performed for a narrow angle range of 0-3◦ or 0-5◦ (in your case). This is for me a justified assumption. However, in the backward region, the extrapolation is based on a polynomic fit and not on any kind of scattering theory. In this case, I can believe that it works well for the very narrow 177-180◦ angle range in the works of Dabrowska et al., 2015 and Escobar-Cerezo et al., 2018. But you applied it for a much broader angle range of 160-180◦, and here I really need some solid proof of this method being justified. The later calculated back-scattering depolarization ratio values cannot be accepted either before your extrapolation is not verified.

Response:

Thank you very much for the constructive comments.

We re-measured scattering matrices for both "pristine loess" and "milled loess" using an improved matrix measurement apparatus covering scattering angles from 5 °to 175 °. Newly measured scattering matrices were in good agreement with measurement results using the previous apparatus in the range of 5-160 °. We also re-constructed synthetic scattering matrices for these two loess samples based on the measurements over 5-175 °, and we think the extrapolated results at 180 °angle are much more rigorous than before. Based on extrapolated values of $F_{22}/F_{11}$ at 180 °, we re-calculated backscatter depolarization ratios for these two loess samples. At last, we re-updated average scattering matrix for loess dust.

Accordingly, we have modified the related descriptions of apparatus, experimental results, synthetic scattering matrices and average scattering matrix in the revised manuscript, and we also re-drawn Figures 3-6, as shown below. In addition, we re-uploaded measured results to a new dataset, which is available at https://github.com/liujia93/Scattering-matrix-for-loess-dust.

We have made necessary modifications in the revised manuscript:

[Figure]

"Figure 3. Layout diagram of the experimental apparatus after backscattering angle expended."

[Figure]

"**Figure 4.** Measured non-zero scattering matrices for "pristine loess" and "milled loess". It should be noted that "milled loess" is the same sample as the "Luochuan loess" in Liu et al. (2019).""

[Figure]

"**Figure 5.** Synthetic scattering matrices for "milled loess" and "pristine loess". Lines are synthetic matrices and plots are measured values.""

[Figure]

"**Figure 6.** Previous average scattering matrix (green lines and solid circles) (Liu et al., 2019) and updated average scattering matrix (red lines and solid squares) for loess dust. Reddish and green shadows stand for the areas covered by results for different loess samples with or without "pristine loess" included, respectively."

Technical Comments: I did not do any language/technical correction because the manuscript needs a bigger revision.

Response:
   Thank you again for all the valuable comments. We have tried our best to response these comments and modified related descriptions in the revised manuscript. We also have tried our best to correct language mistakes in the manuscript. We hope that you can re-consider our manuscript.

[revised manuscript text omitted]

---

## Author Comment (AC5) · 14 Jun 2020

**Responses to the comments of reviewer 5**

The authors really appreciate the valuable comments and constructive suggestions from the reviewer. The suggestions and comments of reviewer are listed in black font, and responses are highlighted in blue. The changes made in the revised manuscript are marked in red font.

**Comments from reviewer 5:**

In Liu et al. the paper focuses on describing the scattering function of a sample that was collected from the Chinese Loess Plateau and subsequently milled to change the physical properties of the particles. The major conclusion gleaned by the authors in the article is that the size of the particles affects the scattering properties. The paper does describe well the need for the research being performed on complex systems, but systematic experiments need to be performed to start to tease out some of that information instead of broad statements about size since that is what they were trying to control. The authors mention that the size distribution is the major factor, but refractive index and micro structure are not ignorable (line 237-240) and then seem to discount that the shape and refractive index have little effect (line 318). They additionally mention that the Refractive index is different (Table 1), but do not seem to try and account for the difference using any kind of modeling to show that it is primarily size. Or identify as to why these are different for the same material.

Response:
Thank you very much for reviewing our manuscript and all these constructive comments.
In our study, we investigated two loess samples with large difference in their particle size distributions from the perspective of long range transportation of dust, the effective size parameters of these two samples are 580 and 30, respectively. Optical simulation results of Gaussian spheres showed that, for particles with above two size parameter, the effects of size parameter and irregularity on scattering matrix elements are roughly opposite (Liu et al., 2015). And the effect of size was qualitatively confirmed by experimental results of these two loess samples, so we concluded that the difference in size distributions plays a major role in leading to discrepancies in measured scattering matrices. However, for particles with effective size parameter smaller than "pristine loess" (580) but larger than "milled loess" (30), optical simulations of Gaussian spheres showed that both size and irregularity have roughly similar effects on matrix elements $F_{33}/F_{11}$, $F_{34}/F_{11}$ and $F_{44}/F_{11}$, it becomes impossible to identify the main factor of influence. In such cases, only qualitative analyses is not enough anymore, supports and cooperation from optical modeling experts are essential to further explore the reasons of discrepancies in matrix elements. Therefore, we did not investigate samples with more sizes, and only measured scattering matrices for two loess samples with large difference in their size distributions. We very hope that there are optical modeling experts interest in our preliminary experimental results and cooperate with us to investigate more samples, since only combinations of experimental measurements and optical simulations are more meaningful.
As mentioned above, with the assistant of qualitative analyses of simulations of Gaussian spheres, we think difference in size distributions is the main reason for these

discrepancies in measured scattering matrices for loess. However, even though it is hard to exactly quantify the change of particle morphology, its effect on scattering matrix is also obvious (Liu et al., 2015). Furthermore, difference of real part of refractive indices of loess samples may also have effects on scattering matrix (Muinonen et al., 2007). So based on qualitative analyses of optical simulation results, we can only draw conclusions that size distribution plays a major role in leading to different scattering matrices while differences in factors such as refractive index and micro structure have relatively small and recessive contributions..

As for the descriptions of refractive index, there is a clerical error in original manuscript, which should be "larger particles have relatively smaller real part of refractive index". For specific material, the refractive index is inherent and unique. SALD-2300 retrieves refractive index of particles by reproducing measured light intensity distributions based on Mie theory. The retrieved refractive index can be regard as a kind of optical equivalent refractive index, it is close to inherent refractive index of measured material but not necessarily the same.

Kinoshita (2001) retrieved refractive indices for alumina dust (whose inherent refractive index is known to be 1.76) with 1 μm and 5 μm diameter using the same method as SALD-2300, the retrieved real parts were 1.80 and 1.75 respectively, the difference was small but cannot be ignored, and this phenomenon was explained as the effect of nonspherical nature by Kinoshita. In our study, we also found larger particles have smaller real part of refractive index, so we think the difference in real part is real and can be explained using the same reason.

This paper does not show significant new data or a new approach to understanding the optical properties of aerosol particles that had not been published previously by the group. The technique has been described by the authors at least twice previously in prior publications and one of the 2 sample sets is already published elsewhere (Liu et al. 2019 and 2018). The paper itself needs to be edited further and reorganized as there are multiple sections that are very similar but spread out through the paper. This paper appears to be more of an addendum to the Liu et al 2019 article than a stand-alone article. Based on these above points, I would be hesitant to recommend this paper for publication as is since there is little information that is novel and there are some unsubstantiated claims throughout.

Response:
Thanks a lot for your comments, and we also appreciate your attention on our previous works.

As we mentioned in the manuscript, the effect of particle size on scattering matrices of mineral dust is still not clear enough, and there is no published research on the effect of size on scattering matrices for loess. Loess dust originating from Chinese Loess Plateau usually forms sand storms in spring season, affecting its source and downwind regions in East Asia. In this study, we experimentally investigated scattering matrices for loess dust with large discrepancy in the size distributions, which is meaningful from the perspective of dust long range transportation. Measured scattering matrices for both coarse "pristine loess" and fine "milled loess" samples are meaningful for the refinement of shape distributions of widely used spheroids as well as the validation and development of more advanced models (Dubovik et al., 2006; Li et al., 2019; Liu et al., 2015). These models will help to improve the retrieve accuracy of aerosol properties

from remote sensing observations over both dust source regions and downwind remote regions.

Until now, the scattering matrix measurement method, synthetic matrix and average matrix construction method have been rigorous enough. Therefore, we just followed the previous methods, and we think that measurement results of samples with atmospheric implication are more important than the improvement of these methods to some extent.

During the revision of the manuscript, we re-measured scattering matrices for both "pristine loess" and "milled loess" samples using an improved experimental apparatus, the maximum backscatter angle coverage of which was extended from 160 ° to 175 °. Newly measured results were in good agreement with measurement results using the original apparatus in the range of 5-160 °. The extension of scattering angle made the polynomial extrapolation of matrix elements $F_{11}(\theta)/F_{11}(10°)$ and $F_{22}(\theta)/F_{11}(\theta)$ at backscattering angles more rigorous when constructing synthetic matrix, and calculated backscattering depolarization ratios were also more reliable.

We have modified related descriptions about apparatus as well as measured, synthetic and average scattering matrices. We also have re-organized the manuscript and added more descriptions about atmospheric implication. And we hope that you will reconsider our manuscript.

Specific Comments:
Line 26: Please specify what this % is, from written it appears to be total aerosol loading worldwide.

Response:
Thanks a lot for pointing this out. We have specified the meaning of "%" in the revised manuscript:
"During aerosol characterization experiments ACE-Asia, mass balance calculations indicated that 45-82 % of atmospheric aerosol mass at observation sites in China were attributed to Asian dust (Zhang et al., 2003)."

Line 36: Please specify what 'r' refers to specifically.

Response:
Thanks a lot for your comments. The 'r' stands for aerodynamic diameter, which is used to characterize transportation and deposition ability of particle. In our study, we measured optical diameter of dust, because, to our best knowledge, there is no instrument available to measure aerodynamic diameter of particles larger than 20 μm, while measurement range of SALD-2300 covers optical diameter from 0.017-2500 μm.

As for the relationship between aerodynamic diameter and optical diameter, Chen et al. (2011) showed that the ratio of aerodynamic diameter to optical diameter is about 0.94-1.21 for Asian dust. Furthermore, according to Li et al. (2018), the ratio of aerodynamic diameter to optical diameter is about 1.15. In our study, we did not make a strict distinction between these two kinds of diameter, because this did not affect the assessment of whether the two loess samples are capable of long range transportation.

Line 42: Remove "Without a doubt"

Response:
   Thank you for pointing this out. We have remove these words in the revised manuscript:
   "Optical properties of dust particles vary with changes of their size distributions."

Line 55: 'Furthermore ... scattering matrices'. This sentence is not completely coherent and needs to be rewritten.

Response:
   Thanks for your suggestion. We have rewritten this sentence in the revised manuscript:
   "Most published literatures of experimental measurements of scattering matrices focused more on similarities and discrepancies between different kinds of mineral dust, or between the same kinds of dust sampled from different sources. Furthermore, some researches paid more attention to the effect of particle size distribution on scattering matrices."

Line: 99-100: What was the injection type for the laser particle sizer? Were they injected in solution or dry?

Response:
   Thank you for the comments. During the measurement of size distribution, dry loess particles were injected into the measurement unit of laser particle sizer. We have added related descriptions in the revised manuscript:
   "The size distributions of "pristine loess" and "milled loess" were determined by a laser particle sizer (SALD-2300; Shimadzu) using dry measurement method, dry loess particles were injected into the measurement unit of laser particle sizer, and three independent repeated measurements were conducted for each sample."

Line 101: Size comparison can be difficult between the two samples due the fact that the original dust sample has a bimodal distribution. This distribution itself will lead to very different scattering properties, whereas the milled sample is a more uniform size. What is the cause of the bimodal shape? Could this be due to a heterogeneity of mineral types being different sizes and having large differences in scattering properties that are then not comparable to the milled sample?

Response:
   Thanks for your comments. We agree that it is difficult to compare bimodal distribution with unimodal distribution. That is why we employed effective radius and effective standard deviation, with the help of these two parameters, we can compare

particles with different size distributions. Volten et al. (2001) showed that the directly sampled and unprocessed red clay, loess, volcanic ash and Sahara sand have different size distributions. This may be because these samples contain different mineral components, these components have distinct size distributions and finally lead to different size distributions of these samples. In our study, the bimodal distribution of "pristine loess" may also be explained using the same reason, and after ball milling, size distributions of different mineral components tend to be the same unimodal distribution. The difference in size distributions can be reflected by measured scattering matrices to some extent.

We have modified related descriptions in the revised manuscript:

"As can be seen from Figure 1, the size of "pristine loess" shows a distinct bimodal distribution, after ball milling, particle size of "milled loess" becomes a unimodal distribution."

Line 103: It is stated that the majority of the particles are larger than 5 microns, but there is a peak at 3 and 10 microns. Please reword this section because you use a cutoff of 5 microns earlier for local vs. long range transport.

Response:

Thank you very much for the comments. As we mentioned in manuscript, particles with radii larger than 5 μm cannot be transported over long distances. However, this does not mean that all airborne particles over source regions have radii larger than 5 μm, there are still a part of fine particles. In our study, the number fraction of particles with radii larger than 5 μm are more than 70% in "pristine loess" sample, so we think this sample can be used to represent airborne loess dust over source regions. We have modified some descriptions in the revised manuscript:

"From the viewpoint of atmospheric particle transportation, the majority (number fraction more than 70%) of "pristine loess" particles have radii larger than 5 μm with peaks at about 3.9 and 10.7 μm, thus this sample can be used to represent coarse dust that only affect source regions, like Xi'an City (Yan et al., 2015)."

Line 105: Please define the peaks more clearly for both samples, with a peak maximum and additional parameters to describe the spread.

Response:

Thanks a lot for the comments. We tried to use Origin Software to fit the measured size distributions of loess samples, and we found only Lorentz function have relatively good fit results. The Lorentz function can be written as:

$$y = y_0 + \frac{2A}{\pi} \frac{w}{4(x - x_c)^2 + w^2}$$

where $x_c$ is peak center, $A$ is peak area, $w$ is full width at half maximum, and $y_0$ is offset of y-axis. Fitted results for "milled loess" (left panel) showed that the peak center $x_c$ is 0.55 μm and full width at half maximum $w$ is 0.46. And fitted results for "pristine loess" (right panel) showed that the peak centers $x_c$ are 3.87 and 12.05 μm, and full widths at half maximum $w$ are 1.11 and 12.21.

[Figure]

However, we think the fitted results for both loess samples are not satisfactory, especially for small radius values. Therefore, we used the peak radii of measured results only in the revised manuscript rather than the fitted results:

"From the viewpoint of atmospheric particle transportation, the majority (number fraction more than 70%) of "pristine loess" particles have radii larger than 5 μm with peaks at about 3.9 and 10.7 μm, thus this sample can be used to represent coarse dust that only affect source regions, like Xi'an City (Yan et al., 2015). On the other hand, almost all particles of "milled loess" sample have radii smaller than 2 μm with a peak at about 0.55 μm, and can be used as a representative of fine dust that can be transported over long distance and affect regions far away from dust sources."

Line 110/Table 1: Why is there a difference in the refractive index if they are still the same material? Please provide the error associated with the measurements and propagate through the rest of the calculations.

Response:
Thank you for the comments. As we mentioned above, refractive index of particles retrieved by SALD-2300 is optically equivalent value, and it is not necessarily the same as inherent refractive index of measured material.

The reason for the small difference 0.05 in retrieved real parts of refractive index for our loess samples is because of the nonspherical nature of particles, Kinoshita (2001) also found similar phenomenon for alumina dust with different sizes. The smallest available calculation steps of real and imaginary part of refractive index in the retrieval are 0.05 and 0.01, respectively. All three repeat measurements obtained the same refractive indices for both "pristine loess" and "milled loess". We have added necessary descriptions of the retrieval of refractive index in the revised manuscript:

"During size distribution measurements of loess samples, the retrieval ranges of real part $Re(m)$ and imaginary part $Im(m)$ of refractive index were preset as 1.45-1.75 and 0-0.05, respectively (Volten et al., 2001). The smallest calculation steps of $Re(m)$ and $Im(m)$ are 0.05 and 0.01, respectively. As shown in Table 1, the optimal refractive indices are 1.65+0$i$ for "pristine loess" and 1.70+0$i$ for "milled loess", larger particles have relatively smaller real part of refractive index, which is similar to the results of Kinoshita (2001) and is caused by the nonspherical nature of loess dust. Retrieved

refractive index of particles based on measured light intensity distribution is a kind of optically equivalent refractive index, which is close to the inherent refractive index of the measured particles."

Line 120: how are the samples for SEM prepared? Are they impacted on the surface or collected some other way?

Response:
   Thank you very much for the comments. During sampling process, we sprayed particles vertically onto copper grids through airflow, particles impact and attach on the surfaces of copper grids. We have added necessary descriptions in the revised manuscript:
   "Some particles of each loess sample were sprayed into vessels or sprayed onto copper grids for subsequent size distribution measurements or SEM analyses."

Line 129: What is the detection limit of this instrument? You quote down to 0.0001 wt% in Table 2. This is mainly of interest since I do not know the limits of XRF.

Response:
   Thanks a lot for your comments. For the instrument XRF-1800, the detection limit is 0.0001 wt%. We have added descriptions about the detection limit in the revised manuscript:
   "For the purpose of detecting whether the chemical compositions of loess samples were changed, the oxide compositions of samples before and after milling process, that is the "pristine loess" and "milled loess", were determined using a X-ray fluorescence spectrometer (XRF-1800, Shimadzu), the detection limit of which is 0.0001 wt %."

Table 2: add an additional column with the difference between the pristine and milled samples. Also include that the characterization was performed by XRF in the caption.

Response:
   Thank you for the comments. We added repeat measurements of chemical components of our loess samples using XRF-1800, and obtained the experimental errors from three measured results for each sample. Comparisons of chemical components for "pristine loess" and "milled loess" showed that the differences between these two samples are small and negligible when experimental errors were taken into consideration. Therefore, we did not add column of component differences between these two loess samples. But we have added columns of experimental errors to Table 2 in the revised manuscript. In addition, we also have added descriptions of XRF in table caption and modified descriptions of sample differences in the revised manuscript:

"As can be seen in Table 2, the largest change of content occurs for $SiO_2$, but this change is less than 2.5 % and even smaller than the errors between repeat measurements for "pristine loess" sample, and the change of $ZrO_2$ is only about 0.03 %. It can be concluded that the composition differences between these two samples are very small, and milling process has little effect on chemical compositions for loess samples."

"**Table 2.** Chemical components of "pristine loess" and "milled loess" measured by XRF-1800."

| Components | Pristine loess (wt %) | Pristine loess error (wt %) | Milled loess (wt %) | Milled loess error (wt %) |
|---|---|---|---|---|
| $SiO_2$ | 63.8278 | 3.0237 | 66.2128 | 2.0900 |
| $Al_2O_3$ | 12.3091 | 0.3772 | 11.6487 | 0.2018 |
| $CaO$ | 9.2943 | 0.9455 | 7.8286 | 0.6450 |
| $Fe_2O_3$ | 5.5260 | 0.8817 | 5.6390 | 0.7411 |
| $K_2O$ | 3.3971 | 0.3004 | 3.3574 | 0.2358 |
| $MgO$ | 2.7536 | 0.4522 | 2.4843 | 0.2665 |
| $Na_2O$ | 1.2802 | 0.0243 | 1.3470 | 0.0214 |
| $TiO_2$ | 0.8017 | 0.0595 | 0.7939 | 0.0579 |
| $P_2O_5$ | 0.3340 | 0.0452 | 0.2549 | 0.0018 |
| $SO_3$ | 0.2370 | 0.1056 | 0.1687 | 0.0721 |
| $MnO$ | 0.1240 | 0.0294 | 0.1196 | 0.0120 |
| $ZrO_2$ | 0.0583 | 0.0104 | 0.0846 | 0.0122 |
| $SrO$ | 0.0348 | 0.0064 | 0.0299 | 0.0059 |
| $Rb_2O$ | 0.0177 | 0.0041 | 0.0174 | 0.0040 |
| $Co_2O_3$ | NT* | - | 0.0159 | 0.0049 |
| $Y_2O_3$ | NT* | - | 0.0061 | 0.0025 |

Line 124: The aggregated particles are all on the large size of the size distribution, would this affect the scattering properties greatly or are they artefacts from particle collection for SEM analysis?

Response:
Thanks a lot for your valuable comments. We think these aggregated large particles are more likely artefacts of the sampling process using copper grids, since we spay particles onto grids directly. Therefore, we resampled particles for SEM analyses and obtained more representative images for "milled loess".

Figure 2: The SEM image for the pristine loess only shows particles in the 10s of microns, it is not a representative image of what the particles actually would look like since the peaks are at ~3 and 10 microns. Additionally, the image for the 'milled loess'

is the same as previously published in the prior manuscript. Please provide representative and comparative SEM images.

Response:
    Thank you very much for pointing this out and thanks for your attention on our previous work. The peaks of radii of "pristine loess" particles are about 3.9 and 10.7 μm, corresponding particle diameters are about 7.8 and 21.4 μm. We resampled loess particles and performed SEM analyses again. Figure 2 in the revised manuscript has been updated using a more representative image of "pristine loess", particle sizes in which are much closer to peaks measured by laser particle sizer, as well as a new image of "milled loess". In addition, we think optical equivalent diameter measured by laser particle sizer are similar to but not exactly equal to geometric size in SEM images.
    We have updated Figure 2 in the revised manuscript:

[Figure]

[Figure]

    "**Figure 2.** SEM images for "pristine loess" (left panel) and "milled loess" (right panel)."

Line 126: What size ZrO2 ball were used and were they milled wet or dry?

Response:
    Thanks a lot for your comments. The "milled loess" sample was prepared by dry ball milling method, and $ZrO_2$ balls with 6 mm diameter were used. We have added related descriptions in the revised manuscript:
    "During the dry milling process, non-metal grinding balls with 6 mm diameter were used, the main component of which is $ZrO_2$."

Line 153-160: I like that the detectors are defined differently, but it would be better to have a different description that 'monitor' and 'detector' as they are both the same pmt detectors just with different functions.

Response:
    Thank you very much for the comments. We have modified the related descriptions in the revised manuscript:

"A photomultiplier named as the "detector", a 532 nm quarter-wave plate $Q$ as well as a polarizer $A$ are fixed on a rotation arm, rotation center of which is coincides with the center of aerosol nozzle. Before scattered light is detected by the "detector", it successively passes through $Q$ and $A$. The dark cassette used to encapsulate the "detector", $Q$ and $A$ in previous apparatus is removed, which facilitate the adjustment of orientation angles of $Q$ and $A$. The "detector" is controlled by an electric rotary table and is able to scan scattering angles from 5° to 175°. Another photomultiplier named as the "monitor" is fixed at 30° scattering angle to record variations of dust aerosols."

"Fluctuations of dust aerosols can be eliminated by normalizing measurements of the "detector" using $DC(30°)$ measured by the "monitor"."

Figure 3: This does not seem necessary as the technique has been described twice previously.

Response:

Thanks a lot for the comments.

During the revision of the manuscript, we improved experimental apparatus by extending coverage of the maximum backscattering angle from 160° to 175°, and we re-measured scattering matrices for both "pristine loess" and "milled loess" samples. In this way, during the construction of synthetic matrices, the values of matrix elements $F_{11}(160°)/F_{11}(\theta)$ and $F_{22}(\theta)/F_{11}(\theta)$ at exact backscattering angle 180° obtained by extrapolations were more reliable. For the extension of angle coverage of apparatus, mechanical structure of detection part of scattered light was adjusted and optimized. The dark cassette used to encapsulate the "detector", $Q$ and $A$ in previous apparatus is removed, which also facilitate the adjustment of orientation angles of $Q$ and $A$. Therefore, we still showed a simple layout diagram of the improved apparatus in Figure 3 in the revised manuscript, and the photograph of improved apparatus in the following figure and detailed validation results using water droplets had been shown in our another work (Liu et al., 2020).

[Figure]

[Figure]

We have updated Figure 3 in the revised manuscript:

[Figure]

"**Figure 3.** Layout diagram of the experimental apparatus after backscattering angle expended."

Lines: 215-223: this paragraph is in an odd place as it references past tables and figures.

Response:
Thank you very much for the comments. In this paragraph, we summarized the differences in fundamental characteristics of these two loess samples, and attempted to infer the main reason for the discrepancies in measured matrices. This provides general guidance for the analyses of literatures focusing on particle optical modeling in the next paragraph in the manuscript. Therefore, we think it is necessary to keep this paragraph, but we have modified and simplified it in the revised manuscript:

"In this study, several fundamental properties of loess dust samples were characterized for auxiliary analyses. As shown in Table 1, effective radii for "pristine loess" and "milled loess" are 49.40 μm and 2.35 μm, respectively. The real part of refractive index for "pristine loess" is 1.65 and that for "milled loess" is 1.70. Table 2 shows that the changes of chemical components are negligible. Therefore, it is reasonable to suspect that distinctions in angular distributions of measured scattering matrix elements for two loess samples may be mainly caused by different size distributions (effective radii differ by more than 20 times), while differences in other factors such as refractive index and micro structure have relatively small contributions in leading to different scattering matrices."

218: "loess dust become more irregular after milling process" How is this defined? If you are saying that they become more irregular, then you will need to actually do analysis of the particles themselves to show the change in the shape parameters. Based on the images seen, this statement cannot be made.

Response:
Thanks a lot for your valuable comments. We agree that this statement is not rigorous. To our best knowledge, it is still very hard to use several morphological parameters to adequately describe the real morphologies of irregular dust particles. Therefore, we removed the related descriptions in the revised manuscript.

In addition, we think the best way to evaluate the change of particle irregularity at present may be employ shape models with different parameters, which may not be fully representative of true morphology of dust, to reproduce measured scattering matrices for these two loess samples, the best fitted shape distributions can be retrieved, then the change of particle irregularity can be roughly evaluated. For such evaluation, we definitely need to cooperate with optical modeling experts.

Line 241-253: This paragraph could be combined with the conclusion, it is very repetitive.

Response:

Thank you very much for the comments.

In the previous paragraph in the manuscript, we found that optical simulation results of Gaussian spheres with different size parameters can qualitatively explain the measured discrepancies in scattering matrices for our loess samples, effective size parameters of which differ by 20 times, and Gaussian spheres may be promising in simulating scattering matrix for loess dust. In this paragraph, we further tried to use Gaussian spheres to explain differences of scattering matrices for other kinds of particles with different sizes, such as olivine and forsterite, and found that simulation results of Gaussian spheres cannot explain these differences. The reason may be that the effects of micro structure and refractive index become more obvious when the difference in size are relatively small, or it may be that Gaussian spheres cannot be used for other kinds of particles.

Therefore, in order to prevent readers from mistakenly thinking that Gaussian spheres may be universal for optical simulation of different kinds of particles, we think it is necessary to keep this paragraph in the revised manuscript. In addition, since our work focuses on loess dust, so we mentioned these statements here rather than in the Conclusions section. According to the comments, we have removed and modified repetitive descriptions in this paragraph in the revised manuscript:

"In this work, a relatively good case is presented to show the effect of size distribution of loess dust on scattering matrices because effective radii of "pristine loess" and "milled loess" differ by more than 20 times. The influence of loess particle size is roughly verified through qualitative analyses of simulation results of Gaussian sphere, which deepen the understanding of this effect. For more detailed explanations, quantitative analyses are still needed based on much more optical simulations of Gaussian spheres. However, besides size distribution, physical properties such as refractive index and micro structure also play important roles in determining scattering matrices of dust particles. When the difference in particle size distributions or effective radii is relative small, the influences of other factors may become dominant or un-ignorable. This may be the reason why the effect of size distribution on measured scattering matrices for olivine samples cannot be concluded clearly (Mu ñoz et al., 2000). And this may also be the reason why effective radii cannot be used to explain all the

discrepancies in matrix elements for forsterite samples based on simulation results of Gaussian spheres (Volten et al., 2006b). Another reason may be that Gaussian spheres are not suitable models to reproduce scattering matrix for forsterite dust, as optical modelling of irregular mineral dust is still a challenging subject."

Figure 4/5: Could these be combined? You could have the synthetic scattering matrix as a different color and a line. It took me a while to see what the difference was between the 2 figures.

Response:
    Thanks a lot for your comments. We have redrawn the synthetic scattering matrices in Figure 5 and added necessary descriptions in its caption in the revised manuscript:

[Figure]

    "**Figure 4.** Measured non-zero scattering matrices for "pristine loess" and "milled loess". It should be noted that "milled loess" is the same sample as the "Luochuan loess" in Liu et al. (2019)."

[Figure]

"**Figure 5.** Synthetic scattering matrices for "milled loess" and "pristine loess". Lines are synthetic matrices and plots are measured values."

Figure 6: Could you specify all the samples that were used in this figure? Either here or in the text.

Response:
   Thank you very much for the comments. We have added descriptions of the samples used to construct average matrix for loess dust in the revised manuscript:
   "At last, the previously published average scattering matrix for loess, which consists of results for Hungary loess, milled Yangling loess and milled Luochuan loess (the latter two were sampled from CLP), was updated using new sample "pristine loess" from Luochuan, by averaging synthetic matrices for different loess samples."

318-319: "other factors ..." this is misleading, since there was no discussion on how the difference in RI affected the sample and no experiments were performed to single these factors out from the size effect. This is also in contrast to earlier where it is stated in line 239-240 "while other factors are also not ignorable"

Response:
   Thanks a lot for your comments. It is very hard to separate single factor from others. In our study, qualitative analyses of simulation results of Gaussian spheres showed that the difference in sizes can be used to roughly explain these discrepancies in scattering matrices for two loess samples (Liu et al., 2015). Furthermore, analyses of optical simulation results showed that both refractive index and micro structure do affect scattering matrix to some degree, but these two factors cannot be used to explain all the

discrepancies in scattering matrix elements (Liu et al., 2015; Muinonen et al., 2007). Based on limited available literatures focusing on optical simulations, we think that these discrepancies in scattering matrices are mainly caused by differences in size distributions, while differences in factors such as refractive index and micro structure have relatively small and recessive contributions. We have modified the related descriptions in the revised manuscript:

"In summary, different factors have different or similar effects on a certain matrix elements. The discrepancies in scattering matrices for "milled loess" and "pristine loess" can be mainly interpreted from the perspective of difference of effective radii, while differences in other factors such as refractive index and micro structure have relatively small contributions, and Gaussian spheres may be promising models for simulating scattering matrix for loess dust."

"Qualitative analyses of optical simulations of various morphological model showed that the large difference in size distributions (effective radii differ by more than 20 times) caused by milling process plays a major role in leading to discrepancies in scattering matrices for these two samples, while differences in factors such as refractive index and micro structure have relatively small and recessive contributions."

References

Chen, G., Ziemba, L. D., Chu, D. A., Thornhill, K. L., Schuster, G. L., Winstead, E. L., Diskin, G. S., Ferrare, R. A., Burton, S. P., Ismail, S., Kooi, S. A., Omar, A. H., Slusher, D. L., Kleb, M. M., Reid, J. S., Twohy, C. H., Zhang, H., and Kooi, S. A.: Observations of Saharan dust microphysical and optical properties from the Eastern Atlantic during NAMMA airborne field campaign, Atmospheric Chemistry and Physics, 11(2), 723-740, doi:10.5194/acp-11-723-2011, 2011.

Dubovik, O., Sinyuk, A., Lapyonok, T., Holben, B. N., Mishchenko, M., Yang, P., Eck, T. F., Volten, H., Muñoz, O., Veihelmann, B., van der Zande, W. J., Leon, J. F., Sorokin, M., and Slutsker, I.: Application of spheroid models to account for aerosol particle nonsphericity in remote sensing of desert dust, Journal of Geophysical Research: Atmospheres, 111(D11), doi:10.1029/2005JD006619, 2006.

Kinoshita, T.: The method to determine the optimum refractive index parameter in the laser diffraction and scattering method, Advanced Powder Technology, 12.4, 589-602, https://doi.org/10.1163/15685520152756697, 2001.

Li, L., Li, Z., Dubovik, O., Zhang, X., Li, Z., Ma, J., and Wendisch, M.: Effects of the shape distribution of aerosol particles on their volumetric scattering properties and the radiative transfer through the atmosphere that includes polarization, Applied optics, 58(6), 1475-1484, https://doi.org/10.1364/AO.58.001475, 2019.

Li, Z., Wei, Y., Zhang, Y., Xie, Y., Li, L., Li, K., Ma, Y., Sun, X., Zhao, W., and Gu, X.: Retrieval of atmospheric fine particulate density based on merging particle size distribution measurements: Multi-instrument observation and quality control at Shouxian, Journal of Geophysical Research: Atmospheres, 123 ,12,474-12,488. https://doi.org/10.102 9/2018JD028956, 2018.

Liu, J., Yang, P., and Muinonen, K.: Dust-aerosol optical modeling with Gaussian spheres: Combined invariant-imbedding T-matrix and geometric-optics approach, Journal of

Quantitative Spectroscopy and Radiative Transfer, 161, 136-144, http://doi.org/10.1016/j.jqsrt.2015.04.003, 2015.

Liu, J., Zhang, Q., Wang, J., and Zhang, Y.: Light scattering matrix for soot aerosol: Comparisons between experimental measurements and numerical simulations, Journal of Quantitative Spectroscopy and Radiative Transfer, 106946, https://doi.org/10.1016/j.jqsrt.2020.106946, 2020.

Muinonen, K., Zubko, E., Tyynelä, J., Shkuratov, Y. G., and Videen, G.: Light scattering by Gaussian random particles with discrete-dipole approximation, Journal of Quantitative Spectroscopy and Radiative Transfer, 106(1-3), 360-377, doi:10.1016/j.jqsrt.2007.01.049, 2007.

Volten, H., Munoz, O., Rol, E., Haan, J. d., Vassen, W., Hovenier, J., Muinonen, K., and Nousiainen, T.: Scattering matrices of mineral aerosol particles at 441.6 nm and 632.8 nm, Journal of Geophysical Research: Atmospheres, 106(D15), 17375-17401, https://doi.org/10.1029/2001JD900068, 2001.

---

## Author Comment (AC6) · 14 Jun 2020

**Responses to the comments of reviewer 6**

The authors really appreciate the valuable comments and constructive suggestions from the reviewer. The suggestions and comments of reviewer are listed in black font, and responses are highlighted in blue. The changes made in the revised manuscript are marked in red font.

**Comments from reviewer 6:**

General Comments: The manuscript by Liu et al. presents results from the light scattering matrices for the samples collected from Chinese Loess Plateau. Auxiliary analyses including particle size distribution, refractive index, chemical component, and microscopic appearance etc. were also done. Based on their results, the authors conclude that the size distribution play a major role in leading to different matrices. In general, the method developed by the authors is novel and fits the slope of the journal. However, some modifications are necessary before it can be considered for publication. One major comment is that the authors did not discuss the atmospheric implications of this novel method. The authors mentioned that the average scattering matrix changed due to the updated sample "pristine loess" compared to previous studies (Fig. 6), this is very interesting, but how meaningful this is to the atmospheric aerosols study? How accurate will it be if we use this new average scattering matrix in future studies? Also, it is necessary for the authors to ask a native English speaker to review the article.

Response:
   Thank you very much for reviewing our manuscript and all these constructive comments. We have responded to the comments point by point and modified related descriptions in the revised manuscript. We also have invited native English speakers to review our manuscript to correct language mistakes. And we hope that you will reconsider our manuscript.
   Optical modeling of dust particles with non-spherical shapes has been an essential subject. Dubovik et al. (2006) recommended a specific shape distribution of spheroids with different aspect ratios to be used in the retrieval of dust aerosol properties from remote sensing observations, because this shape distribution of spheroids had the best performance in reproducing measured scattering matrices for different kinds of mineral dust published by Volten et al. (2001). Since then, this shape distribution had been widely used in the properties retrieval of airborne dust particles. However, Li et al. (2019) found that shape distributions of spheroids have obvious effects of scattering matrices and further affect radiance distribution and polarization properties of sky light. Therefore, the application of above recommended shape distribution for all kinds of dust with different properties is somewhat unreasonable. Our study provided measurements of scattering matrices of "pristine loess" and "milled loess" with large difference in their size distributions, from the perspective of particle transportation. These measured results are useful for the refinement of shape distributions of spheroids for optical modelling of loess dust, and are finally useful to improve retrieval accuracy of dust aerosol properties over both loess source and downwind regions. Furthermore, the updated average scattering matrix for loess is more representative than before, because it contains measured results of coarse "pristine loess" sample that affects dust

source regions. We think this average scattering matrix is also useful for the verification and development of more advanced morphological models for loess dust than spheroids. The improvement of retrieval accuracy of dust aerosol properties by using shape distributions of spheroids or more advanced models retrieved based on our average scattering matrix still requires specific studies to quantify.

We have added above atmospheric implications in the Introduction and Conclusions in the revised manuscript:

"It is well known that dust particles have distinct non-spherical shapes, thus retrievals of dust aerosol properties, like optical thickness, based on Lorenz-Mie computations will lead to significant errors (Herman et al., 2005; Mishchenko et al., 2003). Optical modeling of dust particles with non-spherical shapes has been an essential subject. Dubovik et al. (2006) employed a mixture of spheroids with different axial ratios as well as spheres to reproduce laboratory measured angular light scattering patterns of dust aerosols presented by Volten et al. (2001), and the best fitted shape distribution of spheroids was obtained and proposed. Subsequent studies on the retrievals of dust aerosol properties from space-based (Dubovik et al., 2011), airborne (Espinosa et al., 2019) and ground-based (Titos et al., 2019) remote sensing observations were all based on this shape distribution. However, the application of a same shape distribution of spheroids for different kinds of dust is somewhat too arbitrary (Li et al., 2019) and may not be suitable for simulating optical properties of loess dust with different size distributions. Furthermore, more precise optical models which are more complex than spheroids and similar to real dust morphology are still needed. Laboratory measurements of angular scattering patterns as well as basic physical features, like size distribution, refractive index and micro structure, of loess dust with different sizes are essential and beneficial to the development of more precise models for loess dust. These models will further useful for more accurate retrievals of dust aerosol properties over both source and downwind regions from remote sensing observations, and more accurate assessments of radiative forcing at different regions."

"In this study, scattering matrices for Chinese loess samples with large difference in their size distributions are investigated. Based on all the measurements, suitable shape distributions of spheroids can be obtained respectively, which are useful for the retrievals of airborne loess dust properties at both source and downwind areas in China or even East Asia. On the other hand, the updated average scattering matrix for loess are meaningful for the validation of exiting models and the development of more advanced morphological models suitable for loess dust, which are also useful to finally improve the retrieval accuracies of dust aerosol properties."

Specific comments:
Pg4, line 100: Figure 1: What is the meaning of r in y-axis? Please explain in the figure caption.

Response:
Thanks a lot for your comments. In Figure 1, x-axis is radius of particle "$r$", and y-axis is normalized number fraction "n($r$)" for particle with radius "$r$". We have redrawn Figure 1 and modified descriptions in figure caption in the revised manuscript:

[Figure]

"**Figure 1.** Normalized number size distributions n($r$) of "pristine loess" and "milled loess". Radius $r$ is plotted in logarithmic scale, and error bars are small and covered by symbols."

Pg6, Fig. 3 is not so clear. Please draw a schematic of the experimental setup.

Response:
Thank you very much for the comments. The schematic of our experimental apparatus can be seen below, and it had been published in our previous paper (Liu et al., 2018), so we did not add it to the revised manuscript.

[Figure]

During the revision of manuscript, we improved the experimental apparatus by extending its maximum coverage of backscattering angle from 160 ° to 175 °, and we re-measured scattering matrices for these two loess samples. In this way, during the construction of synthetic matrices, extrapolated values of scattering matrix elements $F_{11}(\theta)/F_{11}(10°)$ and $F_{22}(\theta)/F_{11}(\theta)$ at 180 ° scattering angle were more reliable. For the extension of angle coverage of apparatus, mechanical structures of scattered light detection part were adjusted and optimized. The dark cassette used to encapsulate the "detector", $Q$ and $A$ in previous apparatus is removed, which also facilitate the adjustment of orientation angles of $Q$ and $A$. We have updated Figure 3 in the revised manuscript using a simple layout diagram of the improved apparatus, and the photograph of apparatus in the following figure had been shown in our another work (Liu et al., 2020).

[Figure]

We also have modified descriptions about Figure 3 in the revised manuscript:

"Figure 3 shows a layout diagram of the improved scattering matrix measurement apparatus."

[Figure]

"**Figure 3.** Layout diagram of the experimental apparatus after backscattering angle expended."

Pg6, line 155: How do you inject the dust aerosols into the setup? Please clarify.

Response:

Thanks a lot for pointing this out. We have added more detailed descriptions in the revised manuscript:

"Subsequently, the modulated incident light is scattered by particles in the scattering zone, which are dispersed using an aerosol generator and are sprayed upwards to scattering zone through a nozzle."

"A dust generator (RBG 1000; Palas) was applied to disperse loess particles (Liu et al., 2018). Re-aerosolized dust aerosols were transported to scattering matrix measurement apparatus using conductive tube and sprayed upwards to scattering zone through nozzle."

Pg8, line 222: The last sentence "while other. . . " is not clear.

Response:

Thank you very much for the comments. We have modified related descriptions in the revised manuscript:

"Therefore, it is reasonable to suspect that distinctions in angular distributions of measured scattering matrix elements for two loess samples may be mainly caused by different size distributions (effective radii differ by more than 20 times), while differences in other factors such as refractive index and micro structure have relatively small contributions in leading to different scattering matrices."

Pg11, line 313-314: The authors should indicate what are these small "milled loess" compared to.

Response:
   Thanks a lot for pointing this out. We have modified related descriptions in the revised manuscript:
   "More specifically, for small "milled loess", relative phase function $F_{11}(\theta)/F_{11}(10°)$ as well as ratios $-F_{12}(\theta)/F_{11}(\theta)$ and $F_{22}(\theta)/F_{11}(\theta)$ are smaller than that for coarse "pristine loess", while ratios $F_{33}(\theta)/F_{11}(\theta)$, $F_{34}(\theta)/F_{11}(\theta)$ and $F_{44}(\theta)/F_{11}(\theta)$ are larger than that for coarse "pristine loess"."

Pg10-11, In the conclusion section, the authors should explicitly explain what is "novel" in the new average scattering matrix compared to their previous study, and the significance of this study.

Response:
   Thank you very much for the valuable comments. We have added related descriptions in the revised manuscript:
   "Synthetic scattering matrices for both "pristine loess" and "milled loess" were defined over 0°-180° scattering angle, and the previously presented average scattering matrix for loess was updated with new coarse "pristine loess" sample included. The phase function $F_{11}(\theta)$ in updated average matrix has larger forward scattering peaks and smaller values at side and backward scattering angles than that in previous average matrix. Compared to previous average matrix, updated average matrix has larger -$F_{12}(\theta)/F_{11}(\theta)$ at side scattering angles, has smaller $F_{33}(\theta)/F_{11}(\theta)$ and $F_{44}(\theta)/F_{11}(\theta)$ at backscattering angles. $F_{22}(\theta)/F_{11}(\theta)$ experiences the largest change before and after update, whose values are enlarged at almost all scattering angles."
   "In this study, scattering matrices for Chinese loess samples with large difference in their size distributions are investigated. Based on all the measurements, suitable shape distributions of spheroids can be obtained respectively, which are useful for the retrievals of airborne loess dust properties at both source and downwind areas in China or even East Asia. On the other hand, the updated average scattering matrix for loess are meaningful for the validation of exiting models and the development of more advanced morphological models suitable for loess dust, which are also useful to finally improve the retrieval accuracies of dust aerosol properties."

References

Dubovik, O., Sinyuk, A., Lapyonok, T., Holben, B. N., Mishchenko, M., Yang, P., Eck, T. F., Volten, H., Muñoz, O., Veihelmann, B., van der Zande, W. J., Leon, J. F., Sorokin, M., and Slutsker, I.: Application of spheroid models to account for aerosol particle nonsphericity in remote sensing of desert dust, Journal of Geophysical Research: Atmospheres, 111(D11), doi:10.1029/2005JD006619, 2006.

Li, L., Li, Z., Dubovik, O., Zhang, X., Li, Z., Ma, J., and Wendisch, M.: Effects of the shape distribution of aerosol particles on their volumetric scattering properties and the radiative transfer through the atmosphere that includes polarization, Applied optics, 58(6), 1475-1484, https://doi.org/10.1364/AO.58.001475, 2019.

Liu, J., Zhang, Q., Wang, J., and Zhang, Y.: Light scattering matrix for soot aerosol: Comparisons between experimental measurements and numerical simulations, Journal of Quantitative Spectroscopy and Radiative Transfer, 106946, https://doi.org/10.1016/j.jqsrt.2020.106946, 2020.

Liu, J., Zhang, Y., Zhang, Q., and Wang, J.: Scattering Matrix for Typical Urban Anthropogenic Origin Cement Dust and Discrimination of Representative Atmospheric Particulates, Journal of Geophysical Research: Atmospheres, 123(6), 3159-3174, https://doi.org/10.1002/2018JD028288, 2018.

Volten, H., Munoz, O., Rol, E., Haan, J. d., Vassen, W., Hovenier, J., Muinonen, K., and Nousiainen, T.: Scattering matrices of mineral aerosol particles at 441.6 nm and 632.8 nm, Journal of Geophysical Research: Atmospheres, 106(D15), 17375-17401, https://doi.org/10.1029/2001JD900068, 2001.